# Optimal Algorithms for Augmented Testing of Discrete Distributions

**Maryam Aliakbarpour**
Rice University
Houston, TX
`maryama@rice.edu`

**Piotr Indyk**
MIT
Cambridge, MA
`indyk@mit.edu`

**Ronitt Rubinfeld**
MIT
Cambridge, MA
`ronitt@csail.mit.edu`

**Sandeep Silwal**
UW-Madison
Madison, WI
`silwal@cs.wisc.edu`

## Abstract

We consider the problem of hypothesis testing for discrete distributions. In the standard model, where we have sample access to an underlying distribution $p$, extensive research has established optimal bounds for uniformity testing, identity testing (goodness of fit), and closeness testing (equivalence or two-sample testing). We explore these problems in a setting where a predicted data distribution, possibly derived from historical data or predictive machine learning models, is available. We demonstrate that such a predictor can indeed reduce the number of samples required for all three property testing tasks. The reduction in sample complexity depends directly on the predictor's quality, measured by its total variation distance from $p$. A key advantage of our algorithms is their adaptability to the precision of the prediction. Specifically, our algorithms can self-adjust their sample complexity based on the accuracy of the available prediction, operating without any prior knowledge of the estimation's accuracy (i.e. they are consistent). Additionally, we never use more samples than the standard approaches require, even if the predictions provide no meaningful information (i.e. they are also robust). We provide lower bounds to indicate that the improvements in sample complexity achieved by our algorithms are information-theoretically optimal. Furthermore, experimental results show that the performance of our algorithms on real data significantly exceeds our worst-case guarantees for sample complexity, demonstrating the practicality of our approach.

## 1 Introduction

Property testing of distributions is a fundamental task that lies at the heart of many scientific endeavors: Given sample access to an underlying unknown distribution $p$, the goal is to infer whether $p$ has a certain property or it is $\epsilon$-far from any distribution that has the property (in some reasonable notion of distance, such as total variation distance) with as few samples as possible. Over the past century [66], this problem has been extensively explored in statistics, machine learning, and theoretical computer science. Indeed, distribution testing (also called hypothesis testing) is now a major pillar of modern learning theory and algorithmic statistics, with applications in learning mixtures of distributions such as Gaussians, Poisson Binomial Distributions and robust learning [40, 45, 36, 75, 34, 45]. The framework has also been extensively studied under privacy [56, 14, 27, 18, 26, 3, 72, 50] and low-memory constraints [8].

38th Conference on Neural Information Processing Systems (NeurIPS 2024).

One of the most natural and well-studied questions in this framework is: given sample access to an unknown distribution $p$, can we determine whether $p$ is equal to another distribution $q$, or $\epsilon$-far from it (e.g. in total variation distance)? This problem has been studied under various assumptions about how we access $q$: It is called *uniformity testing* when $q$ is a uniform distribution, *identity testing* (or "goodness-of-fit") when a description of $q$ is known, and *closeness testing* ("two-sample testing" or "equivalence testing") when we only have sample access to $q$. The primary goal in solving these tasks is to design algorithms that use as few samples as possible. Optimal sample complexity bounds have been established for discrete distributions $p$ and $q$ over a domain of size $n$: $\Theta(\sqrt{n}/\epsilon^2)$ samples for uniformity testing [52, 15, 69, 38] and identity testing [76, 5, 39], and $\Theta\left(n^{2/3}/\epsilon^{4/3} + \sqrt{n}/\epsilon^2\right)$ samples for closeness testing [16, 30, 38, 39]. Other related versions of uniformity, identity, and closeness testing are presented in [63, 6, 19, 51, 46, 74, 12, 13, 68].

Given that the aforementioned results are tight and cannot be improved, any further progress requires equipping the algorithm with additional functionality. A natural approach is to leverage the fact that in numerous applications, the underlying distribution is not completely unknown; some prediction of the underlying distribution may be available or can be learned via a predictive machine learning model. For example, if distributions evolve slowly over time, earlier iterations can serve as approximations for later ones, e.g., network traffic data and search engine queries. Such estimations can often be learned from "older" data by using it to train a predictor or regressor. In linguistics and text processing tasks that involve distributions over words, the length of a word can approximate its frequency, since longer words are known to be less frequent. Another example is when data is available at different "scales." For instance,demographic data on loan defaults at the national level could be informative for data from specific areas.

One challenge to using such information is that it rarely comes with a guarantee of precision. This fact leads to the information being deemed unreliable, as it may poorly predict the underlying distribution. For example, while the national loan default rate might be close to that of a typical area code, it could differ significantly in affluent areas. Thus a natural algorithmic question that arises is how to design algorithms that can exploit predictive information as much as possible without any prior assumptions about its accuracy. The goal is then to design an algorithm that solves the problem with as few samples as possible, given the quality of the prediction.

In this work, we study the fundamental problems of uniformity, identity, and closeness testing in the setting where a prediction of the underlying distribution is provided. This prediction can be formalized by assuming that the distribution testing algorithm has access to a predicted distribution $\hat{p}$ of $p$[1]. This is in addition to having sample access to $p$ as in the standard model (without access to prediction). Our algorithms achieve the *optimal* reduction in the number of samples used compared to the standard case, where the improvement depends on the quality of the predictor $\hat{p}$ in terms of its total variation distance from $p$. Our algorithms can also self-adjust their sample complexity to the accuracy of $\hat{p}$, minimizing sample complexity wherever feasible, without prior knowledge of $\hat{p}$'s accuracy. Our approach ensures that the algorithm is resilient to inaccuracies in predictions and does not exceed the optimal sampling bound in the standard model, even when $\hat{p}$ significantly deviates from the actual $p$. Furthermore, our matching lower bounds demonstrate the optimality of our algorithms. Experimental results additionally confirm the practicality of our algorithms.

**Measuring accuracy of predictor.** We use the total variation distance between the prediction $\hat{p}$ and the unknown distribution $p$ as our measure of predictor accuracy. Previous work often assumed a strong element-wise guarantees, where $\hat{p}_i$ is within a constant multiplicative factor of $p_i$ for *all* domain elements $i$, a constraint that becomes limiting especially for small $p_i$ (see Section 1.3). This paper is the first to study a notion of *average* error between $p$ and $\hat{p}$, measuring via the TV distance. This metric was chosen for its prevalent use in statistical inference and its intuitive interpretation.

## 1.1 Our approach and problem formulation

Our approach to solving a distribution testing problem (uniformity, identity, or closeness testing) consists of two components: *search* and *test*. At a high level, *search* aims to guess $\|p - \hat{p}\|_{\mathrm{TV}}$, and *test* performs the actual distribution testing using the guess of the accuracy provided by *search* as a suggested accuracy level.

---

[1]We assume we know all of the probability values of the prediction $\hat{p}$ at all domain elements.

More precisely, our augmented *test* component aims to evaluate whether $p = q$, while receiving $\hat{p}$ and a *suggested* accuracy level $\alpha$ (which may or may not reflect the true distance between $p$ and $\hat{p}$). In addition to two possible outcomes of accept and reject in the standard setting[2], our augmented *test* component may output inaccurate information when it determines that the $\hat{p}$ is not $\alpha$-close to $p$. Our requirements for the augmented *test* component are the following: $i$) If the test is conclusive, i.e., it chooses to output accept or reject, the answer should be correct regardless of $\hat{p}$'s accuracy. $ii$) If $\hat{p}$ is indeed $\alpha$-close to $p$, the *test* component should not output inaccurate information.

We emphasize that the guess $\alpha$ *is not guaranteed* to be correct or may not even be a valid upper bound on the true TV distance. Thus, the algorithm is afforded a degree of flexibility to forego solving the problem when the distributions are not within $\alpha$ proximity (and can try again with a new guess).

Our *search* component aims to identify an appropriate accuracy level $\alpha$ such that the *test* component can test in a conclusive manner by returning accept or reject. Since the true value distance $\|p - \hat{p}\|_{\mathrm{TV}}$ is not known, we start by guessing a small $\alpha$, corresponding to the most accurate $\hat{p}$ and the fewest samples needed, run the augmented *test* component with this $\alpha$ and $\hat{p}$, and verify the conclusiveness of the testing. If inconclusive, our guess $\alpha$ is increased to a level that we can afford testing by doubling the sample size, and the *search* component proceeds with the next $\alpha$. It continues until the desired accuracy is reached, and accept or reject is returned. Then, *search* halts with that result.

Clearly, if the accuracy level guess $\alpha$ is at least $\|p - \hat{p}\|_{\mathrm{TV}}$, the *test* component has to output accept or reject with high probability. Thus, we show that it is unlikely that *search* proceeds when $\alpha \gg \|p - \hat{p}\|_{\mathrm{TV}}$. Therefore, this method does not significantly increase the sample complexity, potentially increasing it by at most an $O(\log\log(n/\epsilon))$ factor in expectation. The *search* component is applicable to all of the problems we study and we defer all details of the *search* component to Section B. The remainder of the main body focuses on designing the augmented testers, i.e. the *test* component.

**Definition 1.1** (Augmented tester)**.** *Suppose we are given four parameters, $\alpha \in [0, 1]$, $\epsilon \in (0, 1)$, $\delta \in (0, 1)$, $n \in \mathbb{N}$, and two underlying distributions $p$ and $q$, along with a prediction distribution $\hat{p}$ over $[n]$. Suppose $\mathcal{A}$ is an algorithm that receives all these parameters, and the description of $\hat{p}$ as input, and it has sample access to $p$ and $q$. We say algorithm $\mathcal{A}$ is an $(\alpha, \epsilon, \delta)$-augmented tester for closeness testing if the following holds for every $p$, $q$, and $\hat{p}$ over $[n]$:*

- *If $\hat{p}$ and $p$ are $\alpha$-close in total variation distance, the algorithm outputs* inaccurate informa-tion *with a probability at most $\delta/2$.*

- *If $p = q$, then the algorithm outputs* reject *with a probability at most $\delta/2$.*

- *If $p$ is $\epsilon$-far from $q$, then the algorithm outputs* accept *with a probability at most $\delta/2$.*

*In this definition, if the description of $q$ is known to the algorithm instead of having sample access, we say $\mathcal{A}$ is an $(\alpha, \epsilon, \delta)$-augmented tester for identity testing. If $q$ is a uniform distribution over $[n]$, then we say $\mathcal{A}$ is an $(\alpha, \epsilon, \delta)$-augmented tester for uniformity testing.*

To highlight the distinction between this definition and the standard definition, note that in the standard regime, no prediction $\hat{p}$ is available to the algorithm, and the algorithm lacks the option of outputting inaccurate information.

**Remark 1.** *We assume that $\delta$ is a small constant, but, a standard amplification technique via Chernoff bounds can achieve an arbitrarily small confidence parameter $\delta$ with a $O(\log(1/\delta))$ overhead.*

## 1.2 Our results

**Our theoretical results:** In this paper we demonstrate that predictions can indeed reduce the number of samples needed to solve the three aforementioned testing problems. Our algorithms are parameterized by both $\epsilon$ and $\alpha$. We provide *tight* sample complexity results (matching upper and lower bounds) for these problems. The sample complexity drops drastically compared to the standard case, depending on the total variation distance between $q$ and $\hat{p}$. Our algorithms are also robust: if the prediction error is high, our algorithm succeeds by using (asymptotically) the same number of samples as the standard setting without predictions.

---

[2]If $p = q$, the standard tester must output accept with high probability. And, if $p$ is $\epsilon$-far from $q$, the standard tester must output reject with high probability. If $p \neq q$, but is $\epsilon$-close to $q$, either answer is considered correct.

**Theorem 2** (Informal version of Theorem 7). *Augmented uniformity and identity testing for distributions over $[n]$, with parameters $\alpha$, $\epsilon$, and $\delta = 2/3$, require the following number of samples:*

$$
s = \begin{cases} \Theta\left(\frac{\sqrt{n}}{\epsilon^2}\right) & \text{if } d \leq \alpha \\[2ex] \Theta\left(\min\left(\frac{1}{(d-\alpha)^2}, \frac{\sqrt{n}}{\epsilon^2}\right)\right) & \text{if } d > \alpha \end{cases},
$$

*where $d = \|q - \hat{p}\|_{TV}$ (q is the known distribution for identity testing, or the uniform distribution for uniformity).*

**Remark 3.** *Note that $d$ is not an input parameter to the algorithm. However, $d$ is determined by $\hat{p}$ and $q$, which are known to the algorithm, allowing us to compute $d$.*

For closeness testing, we prove the following.

**Theorem 4** (Informal version of Theorem 9). *Augmented closeness testing for distributions over $[n]$, with parameters $\alpha$, $\epsilon$, and $\delta = 2/3$, requires $\Theta(n^{2/3}\alpha^{1/3}/\epsilon^{4/3} + \sqrt{n}/\epsilon^2)$ samples.*

It can be seen that e.g., for closeness testing, our non-trivial predictor improves over the best possible sampling bound in the standard model. Specifically, as long as $\alpha = o(1)$, our bound in the augmented model improves over the prior work. At the same time, our algorithms are resilient: even if $\alpha = \Theta(1)$, our sampling bounds do not exceed those in the standard model. Note that all theorems are complemented by *tight* lower bounds. We highlight that all of our algorithms are also computationally efficient, running in polynomial time with respect to $n$ and $1/\epsilon$. The results are summarized in Table 1.

Table 1: Optimal sample complexity bounds in the standard model versus the augmented bounds. $\alpha$ is the suggested accuracy level for the $L_1$-distance between $p$ and $\hat{p}$. $d$ denotes the total variation distance between the known distribution $q$ and $\hat{p}$.

| Property | Standard sample complexity | Augmented sample complexity (this paper) |
|---|---|---|
| Closeness | $\Theta\left(\frac{n^{2/3}}{\epsilon^{4/3}} + \frac{\sqrt{n}}{\epsilon^2}\right)$ [30, 69] | $\Theta\left(\frac{n^{2/3}\alpha^{1/3}}{\epsilon^{4/3}} + \frac{\sqrt{n}}{\epsilon^2}\right)$ |
| Identity/Uniformity | $\Theta\left(\frac{\sqrt{n}}{\epsilon^2}\right)$ [69, 5] | $\Theta\left(\frac{\sqrt{n}}{\epsilon^2}\right)$, if $d \leq \alpha$ $\Theta\left(\min\left(\frac{1}{(d-\alpha)^2}, \frac{\sqrt{n}}{\epsilon^2}\right)\right)$, if $d > \alpha$ |

**Our empirical Results:** We empirically evaluate our augmented closeness testing algorithm on synthetic and real distributions and refer all details to Section E.

As a summary, our algorithm can indeed leverage predictions to obtain significantly improved sample complexity over the SOTA approach without predictions, as well as SOTA algorithms needing very accurate predictions [28].

For distributions similar to our lower bound instances, our augmented algorithm achieves >**20x** reduction in sample complexity to obtain comparable accuracy as the standard un-augmented algorithm, as shown in Figure 1. On real distributions curated from network traffic data, we see sample complexity reductions of up to $40\%$. Furthermore, our algorithm is empirically robust to noisy predictions, in contrast to prior state of the art approaches which assume very accurate, point-wise predictions (CRS'15 in Figure 1).

It is worth noting that our experiments on network traffic data reveal that the actual sample complexity is much lower than the anticipated worst-case sample complexity of our algorithm. In particular, this holds

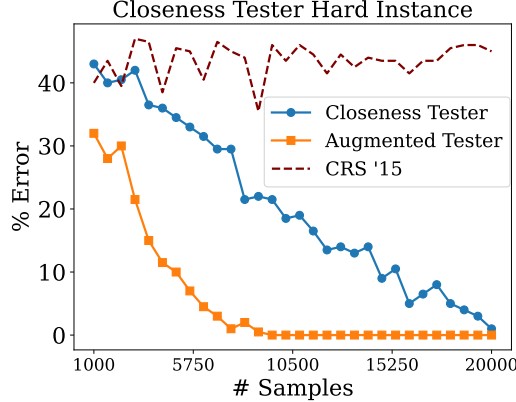

Figure 1: Error vs sample complexity for the theoretically hard instance (See Sec. E).

even when $\hat{p}$ is far from $p$ in terms of total variation
distance. Empirically, if $\hat{p}$ accurately reflects the high-probability elements in $p$, our algorithm can significantly reduce the sample complexity needed for testing by utilizing these heavy hitter "hints" from $\hat{p}$. This is validated by our results and depicted in Figures 5 and 7(b).

## 1.3 Related works

To the best of our knowledge, there have been only three prior works that studied any distribution property testing algorithms with predictions, each assuming a much stronger prediction model:

- Distribution testing with *perfect* predictors [29]: this work studied distribution testing problems, including closeness, identity and uniformity, assuming *query access* to a perfect predictor, i.e., $\hat{p} = p$. They show that, given a perfect predictor, it is possible to obtain highly efficient testers for a wide variety of problems with a small number of queries to the prediction. Unfortunately, the perfect-predictor assumption is often too strong to hold in practice, as demonstrated e.g., in [48] and in our experiments.

- Distribution testing with $(1 + \epsilon)$-*approximate* predictors [28, 67]: these works relax the assumption used in the above paper, requiring only that $\hat{p}_i = (1 \pm \epsilon/2)p_i$ for all $i$ and sufficiently small $\epsilon > 0$. However, this assumption is still quite restrictive, especially for low values of $\epsilon$. Indeed in our experimental setting, such algorithms are not robust to prediction errors (see Section E).

- Support estimation with $c$-approximate predictors [48]: this work focused on the single problem of support estimation, i.e., estimating the fraction of coordinates $i$ such that $p_i > 0$. It further relaxes the assumption in [67] by allowing the predicted probabilities $\hat{p}_i$ to be within a factor of $c$ of the true probabilities $p_i$, for *any* constant approximation factor $c > 1$. This algorithm has been shown to work well in practice. However, the techniques presented in that paper seem to be applicable exclusively to support estimation. Furthermore, their result provably does not hold under the assumption that $p$ and $\hat{p}$ are close in TV distance, as in this paper.

In summary, prior results required either highly restrictive assumptions, or were applicable to only a single problem. None of the previous algorithms worked under the TV distance assumption used in this paper which is arguably the most natural. Further exploration of measures such as $L_p$-distances, KL-divergence, and Hellinger distance are interesting open questions.

We remark that we can mathematically show that these alternative oracles yield provably more power compared to ours (i.e., we make weaker assumptions about the predictor). We provide an alternative to our upper bound techniques for these alternate prediction models in Section G. We demonstrate how a variant of our algorithm, in conjunction with these stronger predictions, implies that uniformity testing, identity testing, and closeness testing in these models can be conducted using only $O(\sqrt{n}/\epsilon^2)$ samples. This low sample complexity effectively circumvents our lower bound for closeness testing, suggesting that these models provide stronger, and arguably less realistic, predictions.

**Other related works:** Other related works discussing algorithms with predictions and general distribution testing are discussed in Appendix A.

## 1.4 Notation and organization

**Notation:** We use $[n]$ to denote the set $\{1, 2, \ldots, n\}$. All of our distributions will be over the domain $[n]$, and we assume $n$ is always known. For a distribution $p$, we denote the probability of the $i$-th element by $p_i$. For any subset of the domain $S \subseteq [n]$ we denote the probability mass of $S$ according to $p$ by $p(S)$. We use $p^{\otimes s}$ to denote the probability distribution of $s$ i.i.d. samples drawn from $p$. We say $p$ is a *known* distribution, if we have access to every $p_i$. We say $p$ is an *unknown* distribution if we have only sample access to $p$. We use $\mathbf{Poi}(\lambda)$ to denote a random variable from a Poisson distribution with mean $\lambda$. Similarly, $\mathbf{Ber}(\lambda)$ indicates a random variable from the Bernoulli distribution that is one with probability $\lambda$. Given two distributions $p$ and $q$ over a domain $\mathcal{X}$, we use $\|p - q\|_{\mathrm{TV}} := \sup_{S \subseteq \mathcal{X}} |p(S) - q(S)|$ to denote the total variation distance between $p$ and $q$. We say $p$ and $q$ are $\epsilon$-close ($\epsilon$-far), if the total variation distance between $p$ and $q$ is at most $\epsilon$ (larger than $\epsilon$).

**Organization:** We provide an overview of our theoretical results in Section 2. The upper bound on augmented closeness testing is presented in Section 3. The search algorithm is detailed in Section B.

Augmented uniformity and identity testing are discussed in Section C, where the upper bounds are presented in Section C.1, and the lower bounds are provided in Sections C.2 and C.3.

Augmented closeness testing is discussed in Section D, with the lower bounds provided in Section D.1. Our empirical evaluations are presented in Section E.

## 2 Overview of our proofs

**Search algorithm:** The search algorithm seeks to find the smallest value of $\alpha$ for which the problem is solvable via the augmented tester. It starts with the lowest value of $\alpha$ (most accurate prediction). Then, it iteratively increases the sample budget across rounds. In each round $i$, it selects an $\alpha_i$ for the augmented tester, $\mathcal{A}$, ensuring operation within the current sample budget. If $\mathcal{A}$ outputs accept or reject, the algorithm echoes this outcome. If inaccurate information is returned, the sample budget doubles for the next round. It is worth noting that this search scheme is applicable to a general distribution testing algorithm with polynomial sample complexity. The algorithm's pseudocode and its performance proof are in Section B.

**Upper bound for augmented identity testing** Let $d$ represent the distance between $\hat{p}$ and $q$ (the prediction and the known distribution). For establishing the upper bound, it is essential to assume $d > \alpha$. If not, the prediction proves unhelpful, and we might as well resort to the standard tester.

Our upper bound relies on a simple but fundamental observation regarding the total variation distance: this distance is the maximum discrepancy between the probability masses that two distributions assign to any subset of their domain. To prove that total variation between two distributions is small entails proving that the discrepancy across every domain subset is small. In contrast, to prove a large total variation distance, one only needs to identify a single subset with a large discrepancy as evidence of large total variation distance. We use the Scheffé set of $\hat{p}$ and $q$—characterized as the collection of elements $x \in [n]$ where $\hat{p}(x) < q(x)$, symbolized by $S$—as evidence of $p$'s divergence from either $q$ or $\hat{p}$. More precisely, it is known that $S$ maximizing the discrepancy between the probability masses of $\hat{p}$ and $q$, implying: $d := \|q - \hat{p}\|_{\text{TV}} = |q(S) - \hat{p}(S)|$ . Next, we estimate the probability of set $S$ according to $p$ with reasonably high accuracy. Given $q(S)$ and $\hat{p}(S)$, then $p(S)$ is either significantly different from $q(S)$, or it deviates from $\hat{p}(S)$. In the first scenario, this discrepancy serves as evidence that $p \neq q$, allowing us to output reject. In the second scenario, the deviation confirms that $p$ is $\alpha$-distant from $\hat{p}$, leading us to output inaccurate information. Further details can be found in Section C.1.

**Lower bound for augmented uniformity testing** We provide two lower bounds for augmented uniformity testing. One is purely based on a reduction to standard uniformity testing for the case where $\alpha \geq d$. (Recall that $d$ was the total variation distance between $q$ and $\hat{p}$). See Section C.2.

The other lower bound applies to the setting where $\alpha < d$. One challenge of this problem was that it is hard to find two difficult distributions that the tester has to distinguish between; usually, for a pair of distributions, we could come up with one valid output that could serve them both. For example, for both uniform distribution, and the famous $\epsilon$-far uniform distribution that assigns probabilities $(1 \pm \epsilon)/n$ to the elements, the algorithm may be able to output inaccurate information. Hence, we cannot draw lower bounds just by asking the algorithm to distinguish between two distributions.

For this reason, we provide three distributions that *look* similar when we draw too few samples. We formalize the similarity of these three distributions using a multivariate coupling argument. We show that these distributions are such that there is no possible answer that is valid for all three of them. Now, (similar to Le Cam's method), suppose we feed the algorithm with samples from one of these three distributions (each with probability 1/3). For any sample set, the algorithm outputs an answer (which may be randomized); however, this answer is considered wrong for at least one of the underlying distributions. This is due to the fact that there is no universally valid answer that is simultaneously correct for all three distributions. Hence, if the algorithm outputs a valid answer with high probability, it must be able to distinguish the underlying distributions to some degree. On the other hand, the indistinguishability result says it is impossible to tell these distributions apart. Thus, we reach a contradiction. And, the lower bound is concluded. See Section C.3 for further details.

**Lower bound for augmented closeness testing:** We provide two separate lower bounds for closeness testing based on the relationship between $\alpha$ and $\epsilon$. Further details are available in Section D.1.

Our first lower bound, as detailed in Theorem 11, employs a reduction strategy from standard closeness testing to augmented closeness testing when $\alpha \geq \epsilon$. This is achieved by taking instances used in standard closeness testing for distributions over $[n-1]$ and embedding them into the first $n-1$ domain elements of a new distribution $p$ defined over $[n]$. The key to this strategy is to put the majority of the distribution's mass ($(1-\alpha)$ mass) on its final element, and we set the prediction $\hat{p}$ to be a singleton distribution over the last element, which is $\alpha$-close to $p$. Clearly, the prediction does not reveal any information about the first $n-1$ elements of $p$, implying that testing the closeness of $p$ in the augmented setting is as challenging as in the standard setting, once the parameters are appropriately scaled.

Our second lower bound, outlined in Theorem 12, is more involved. In the standard setting, the lower bound for closeness testing is derived from the hard instances for uniformity testing with one crucial adjustment: adding new elements with large probability (approximately $n^{-2/3}$) in the distributions. These large elements have identical probability masses in both $p$ and $q$, indicating they do not contribute to the distance between the two distributions. However, their presence in the sample set confuses the algorithm: due to their high probabilities, their behavior in the sample set may misleadingly suggest non-uniformity, complicating the algorithm's task of determining the uniformity of the rest of the distribution. Therefore, the algorithm requires $s \approx n^{2/3}$ samples to first identify these large elements before it can test the uniformity of the remaining distribution. Surprisingly, this requirement of $s \approx n^{2/3}$ samples is significantly higher than the $O(\sqrt{n})$ samples typically sufficient for testing uniformity.

The challenge in our case arises because $\hat{p}$ may disclose the large elements to the algorithm. To establish the lower bound, we set $\hat{p}$ to be the uniform distribution, we generate hard instances of $p$ by adding as many large elements as possible, without altering $\hat{p}$, by keeping the overall probability mass of the large elements limited to $\alpha$. More precisely, $p$ assigns approximately $(1-\alpha)/n$ probability mass to $O(n)$ elements chosen at random, and assigns approximately $n^{-2/3}$ probability mass to $\alpha \cdot n^{2/3}$ elements in the domain. Now, $q$ has two scenarios. Half the time, $q$ is identical to $p$. The other half, $q$ retains the same large elements but deviates slightly from uniformity for the rest of the distribution. Specifically, $q$ assigns probabilities $(1 \pm \Theta(\epsilon))(1-\alpha)/n$ to the randomly chosen $O(n)$ elements, making it $\epsilon$-far from $p$. It is not hard to show that testing closeness of $p$ and $q$ is a symmetric property (since permuting the domain elements does not affect our construction). By leveraging the wishful thinking theorem from [78], we demonstrate that these two scenarios are indistinguishable unless $\Omega(n^{2/3}\alpha^{1/3})$ samples are drawn.

## 3 Upper bound for closeness testing

Our upper bound is based on a technique called *flattening*, which has been previously proposed by Diakonikolas and Kane [42]. This technique is instrumental in reducing the variance of the statistic used for closeness testing by reducing the $\ell_2^2$-norm of the input distributions. We adapt this technique for use in the augmented setting, aiming not only to flatten the distribution based on the samples received from it but also exploiting the prediction distribution $\hat{p}$. We demonstrate that augmented flattening can significantly reduce the $\ell_2^2$-norm of $p$. In our algorithm, we initially check if the $\ell_2^2$-norm of $p$ is reduced after flattening to the desired bound. If not, this indicates that the prediction was not sufficiently accurate, leading us to output *inaccurate*. Conversely, if the $\ell_2^2$-norm of $p$ is small, we proceed with an efficient testing algorithm that requires fewer samples. We describe the standard flattening technique in Section 3.1. Our flattening technique presented in Section 3.2. Finally, we provide our algorithm in Section 3.2.1.

### 3.1 Background on flattening

Suppose we are given $n$ parameters $m_1, m_2, \ldots, m_n$. One can create a randomized mapping $F$ that assigns each $i \in [n]$ to a pair $(i, j)$, where $j$ is drawn uniformly at random from $[m_i]$. Now, consider a given distribution $p$ over $[n]$ and a sample $i$ drawn from $p$. This mapping induces a distribution over pairs $(i, j)$'s in $D := \{(i, j) | i \in [n] \text{ and } j \in [m_i]\}$. We denote this new distribution by $p^{(F)}$ satisfying the relation: $p_{(i,j)}^{(F)} = p_i/m_i$. The core idea of the above mapping is to divide

the probability of the $i$-th element into $m_i$ *buckets*. If the values of $m_i$'s are large for elements in $p$ with higher probabilities, then the resulting distribution $p^{(F)}$ will avoid having any elements with disproportionately large probabilities, thereby naturally lowering its $\ell_2$-norm. Diakonikolas and Kane [42] showed that if we draw $\mathbf{Poi}(t)$ samples from $p$, and set each $m_i$ to be the frequency of element $i$ among these samples plus one, then we have:

$$\mathbf{E}\left[\left\|p^{(F)}\right\|_2^2\right] \leq \frac{1}{t},$$

where the expectation is taken over the randomness of the samples.

**Connection to distribution testing:**  Chan et al. in [30] showed that one can test closeness of two distributions over $[n]$ using $O(n \max(\|p\|_2, \|q\|_2)/\epsilon^2)$ samples. Diakonikolas et al. [42] used this tester in combination of the flattening technique to map distributions to distributions over slightly larger domains with the goal of reducing their $\ell_2$-norms. They have shown that if we use the same mapping to reduce the $\ell_2$-norm of both $p$ and $q$, the $\ell_2$-distance between the two distribution does not change the $\ell_1$ distance between distributions.

**Fact 3.1** ([42])**.** *For any flattening scheme $F$, the $\ell_1$-distance is preserved under $F$. That is, for every pair of distributions $p$ and $q$, we have:*

$$\|p - q\|_1 = \|p^{(F)} - q^{(F)}\|_1.$$

Therefore, to test $p$ and $q$, we can draw samples to come up with the flattening $F$ to reduce the $\ell_2$-norm of one of the underlying distributions. Then, we can test the closeness of $p^{(F)}$ and $q^{(F)}$ with $O(n \max(\|p^{(F)}\|_2, \|q^{(F)}\|_2)/\epsilon^2)$ new samples. Later, Aliakbarpour et al. in [9] showed that $O(n \min(\|p\|_2, \|q\|_2)/\epsilon^2)$ samples is sufficient.

The exact sample complexity is determined by balancing the samples needed to determine the flattening and the samples needed to test closeness of $p^{(F)}$ and $q^{(F)}$. Note that flattened distributions have a larger domain size. Thus, one must also balance the gains obtained from the reduction in $\ell_2$ norm with the increase in the domain.

## 3.2  Augmented flattening

We adapt the flattening argument in [42] to the augmented setting.

**Lemma 3.2.** *Suppose we have an unknown distribution $p = (p_1, p_2, \ldots, p_n)$ over $[n]$ and $\mathbf{Poi}(s_f)$ samples from $p$. Assume we are given a known distribution $\hat{p}$ that is $\alpha$-close to $p$ in TV distance. Then for every $v \leq 1$, there exists a flattening $F$ which increases the domain size by $1/v + s_f$ in expectation and the expected $\ell_2^2$-norm of the $p^{(F)}$ is bounded by:*

$$\mathbf{E}\left[\left\|p^{(F)}\right\|_2^2\right] \leq \frac{2\alpha}{s_f} + 4 \cdot \nu.$$

*The expectation above is taken over the randomness of the samples from $p$.*

*Proof.* Let $\hat{s}_f := \mathbf{Poi}(s_f)$ be a Poisson random variable with mean $s_f$. Let $\mathcal{S}_f$ denote the multiset of $\hat{s}_f$ samples from $p$. Let $f_i$ denotes the frequency of element $i$ in $\mathcal{S}_f$. For every $i$, we define the number of buckets for element $i$ as follows:

$$m_i := \left\lfloor \frac{\hat{p}_i}{\nu} \right\rfloor + f_i + 1. \tag{1}$$

Let $p^{(F)}$ denote the flattened version of $p$ where we split every element $i$ into $m_i$ buckets. We show that the expected $\ell_2$-norm of $p^{(F)}$ is low. Define $\Delta_i$ to be $p_i - \hat{p}_i$. Suppose $A$ is a set of indices $i \in [n]$ for which $\Delta_i \geq \hat{p}_i$. For every $i \in A$, $p_i$ is bounded from above by $2\Delta_i$. On the other hand, for every $i \in [n] \setminus A$, $p_i$ is bounded by $2\hat{p}_i$.

$$\left\|p^{(F)}\right\|_2^2 = \sum_{i \in [n]} \sum_{j \in [m_i]} p^{(F)}(i,j)^2 = \sum_{i \in [n]} \frac{p_i^2}{m_i} = \sum_{i \in A} \frac{2\,\Delta_i \cdot p_i}{m_i} + \sum_{i \in [n] \setminus A} \frac{(2\,\hat{p}_i)^2}{m_i}$$

$$\leq \sum_{i \in A} \frac{2\,\Delta_i \cdot p_i}{f_i + 1} + \sum_{i \in [n] \setminus A} \frac{(2\,\hat{p}_i)^2}{\hat{p}_i / \nu}$$

$$\leq \sum_{i \in A} \frac{2\,\Delta_i \cdot p_i}{f_i + 1} + 4\,\nu.$$

As shown in [42], the expected value of $p_i / (f_i + 1)$ taken over the randomness in the $\mathcal{S}_f$ is bounded by $1/s_f$:

$$\mathbf{E}_F \left[ \frac{p_i}{f_i + 1} \right] \leq \frac{1}{s_f}$$

Therefore, we obtain:

$$\mathbf{E}_F \left[ \left\|p^{(F)}\right\|_2^2 \right] \leq \sum_{i \in A} \frac{2\,\Delta_i}{s_f} + 4\,\nu \leq \frac{2\,\alpha}{s_f} + 4\,\nu.$$

The last inequality above is due to the fact that:

$$\sum_{i \in A} \Delta_i = p(A) - \hat{p}(A) \leq \|p - \hat{p}\|_{\mathrm{TV}} = \alpha. \quad \square$$

### 3.2.1 The algorithm

In this section, we provide an algorithm for testing closeness of two distribution in the augmented setting. Our algorithm receives a suggested accuracy level $\alpha$. Based on $\alpha$, we draw $s_f$ (which depends on $\alpha$) samples from $p$ and use them to flatten $p$. If the resulting distribution has sufficiently small $\ell_2^2$-norm we proceed to the testing phase. Otherwise, we declare that the accuracy level provided is not correct. The pseudocode of our algorithm is provided in Algorithm 1, and we prove its performance in Theorem 5.

---

**Algorithm 1** An augmented tester for testing closeness of $p$ and $q$ with a suggested value $\alpha$

---

1: **procedure** AUGMENTED-CLOSENESS-TESTER($n$, $\alpha$, $\epsilon$, $\delta_1 = 0.1$, $\delta_2 = 2/3$, $\hat{p}$, sample access to $p$ and $q$)
2:      $s_f \leftarrow \min\left(n^{2/3}\alpha^{1/3}/\epsilon^{4/3}, n\right)$
3:      $\hat{s}_f \leftarrow \mathbf{Poi}(s_f)$
4:      **if** $\hat{s}_f > 10\,s_f$ **then**
5:          **return** reject.
6:      $F \leftarrow F(m_1, m_2, \ldots, m_n)$ where $m_i$ are define in Equation 2.
7:      $L_p \leftarrow$ ESTIMATE-$\ell_2^2\left(p^{(F)}, \sum_{i=1}^{n} m_i, 0.05\right)$
8:      $L_q \leftarrow$ ESTIMATE-$\ell_2^2\left(q^{(F)}, \sum_{i=1}^{n} m_i, 0.05\right)$
9:      **if** $L_p > 30\left(\frac{2\alpha}{s_f} + \frac{4}{n}\right)$ **then**
10:          **return** inaccurate information.
11:      **else**
12:          **if** $L_q > 90\left(\frac{2\alpha}{s_f} + \frac{4}{n}\right)$ **then**
13:              **return** reject.
14:          **else**
15:              **return** CLOSENESS-TESTER $\left(\epsilon, \delta = 0.1, b = 90\left(\frac{2\alpha}{s_f} + \frac{4}{n}\right), p^{(F)}, q^{(F)}\right)$.

---

**Theorem 5.** *For any given parameters $n$, $\alpha$, $\epsilon$, and any two unknown distributions $p$ and $q$ and a known predicted distribution $\hat{p}$ for $p$, Algorithm 1 is an $(\alpha, \epsilon, \delta = 0.3)$-augmented tester for testing closeness of $p$ and $q$ which uses $O(n^{2/3}\alpha^{1/3}/\epsilon^{4/3})$ samples.*

*Proof.* We begin the proof by setting our parameters. Set $s_f := \min\left(n^{2/3}\alpha^{1/3}/\epsilon^{4/3}, n\right)$. Suppose we draw $\hat{s}_f = \mathbf{Poi}(s_f)$ samples from $p$. We flatten the distribution $p$ according to the augmented flattening we described in Section 3.2 replacing $\nu$ with $1/n$. We denote the flattening by $F = F(m_1, \ldots, m_n)$ where $m_i$'s are defined in Equation (1):

$$m_i := \lfloor n \cdot \hat{p}_i \rfloor + f_i + 1. \tag{2}$$

In the above definition, $f_i$ is the frequency of element $i$ among the $\hat{s}_f$ drawn samples.

**Proof of correctness:** Suppose we flatten both $p$ and $q$ according to $F$. The algorithm estimates the $\ell_2^2$ of $p^{(F)}$ and $q^{(F)}$. According to Fact F.1, these estimates are within a constant factor of their true values each with probability at least $1 - 0.05$:

$$\frac{\left\|p^{(F)}\right\|_2}{2} \leq L_p \leq \frac{3\left\|p^{(F)}\right\|_2}{2} \qquad \text{and} \qquad \frac{\left\|q^{(F)}\right\|_2}{2} \leq L_q \leq \frac{3\left\|q^{(F)}\right\|_2}{2}. \tag{3}$$

To prove the correctness of the algorithm, we show the three desired property of an augmented tester which are defined in Definition 1.1. First, assume $\hat{p}$ is $\alpha$-close to $p$. We show the probability of outputting inaccurate information is small. Using Lemma 3.2, after applying $F$, the expected $\ell_2^2$-norm of $p^{(F)}$ is bounded by:

$$\mathbf{E}\left[\left\|p^{(F)}\right\|_2\right] \leq \frac{2\alpha}{s_f} + \frac{4}{n}.$$

Using Markov's inequality, with probability at least 0.95, $\ell_2^2$-norm of $p^{(F)}$ is at most 20 times the above bound. Combined by the upper bound for $L_p$ in Equation (3) and the union bound, in this case $L_p$ is bounded by $30 \cdot (2\alpha/s_f + 4/n)$ with probability at least 0.9. Hence, we the algorithm does not output inaccurate information in Line 10 with probability more than 0.1.

Second, we show that if $p = q$, the algorithm outputs reject with small probability. We may output reject in three cases: 1) in Line 5 2) in Line 13 and 3) in Line 15. Using Markov's inequality, a Poisson random variable is more than 10 times its expectation with probability at most 0.1. We show the other two cases are unlikely as long as $L_p$ and $L_q$ are accurate estimates of the $\ell_2^2$s of $p^{(F)}$ and $q^{(F)}$ (which happens with probability of 0.9). If $p = q$, $p^{(F)}$ and $q^{(F)}$ are identical. Therefore, in the algorithm $L_p$ and $L_q$ are the estimations of the $\ell_2^2$-norm of the same distribution. Note that we output reject in Line 13, only when $L_p < 3L_q$. Using Equation (3), this event does not happen unless at least of the estimates are inaccurate (which has a probability at most 0.1). Furthermore if the estimated $\ell_2^2$-norm are accurate, the minimum $\ell_2^2$-norm of $p^{(F)}$ and $q^{(F)}$ are bounded by $90 \cdot (2\alpha/s_f + 4/n)$ implying the tester would work correctly with probability at least 0.9. Using the union bound, the probability of outputting reject is bounded by 0.3.

Third and last, we show that if $p$ is $\epsilon$-far from $q^{(F)}$, the probability of outputting accept is small. We only output accept in Line 15. Similar to the second case we have discussed, as long as the $\ell_2^2$-norm are accurate, the tester does not make a mistake with probability more than 0.1. Therefore, the overall probability of making a mistake is bounded by 0.2.

**Analysis of sample complexity:** For the flattening step we have used $\hat{s}_f \leq 10s_f$. After flattening, the new domain size is bounded from above by:

$$\sum_{i=1}^{n} m_i \leq \sum_{i=1}^{n} n \cdot \hat{p}_i + f_i + 1 = n + \hat{s}_f + n \leq 12n.$$

Using Fact F.1, estimation of $\ell_2^2$-norm requires $O(\sqrt{n})$ samples. Using Fact F.2, the number of samples we use for the test in Line 15 is $O(n \cdot b/\epsilon^2)$ samples. Hence, the total sample complexity is:

$$\# \text{ samples} = O\left(s_f + \sqrt{n} + \frac{n \cdot b}{\epsilon^2}\right) = O\left(s_f + \sqrt{n} + \frac{n \cdot \sqrt{2\alpha/s_f + 4/n}}{\epsilon^2}\right)$$

$$= O\left(\frac{\sqrt{n}}{\epsilon^2} + \frac{n^{2/3}\alpha^{1/3}}{\epsilon^{4/3}}\right).$$

In the last line we use that $s_f := \min\left(n^{2/3}\alpha^{1/3}/\epsilon^{4/3}, n\right)$. Thus, the proof is complete. $\qquad \square$

## Acknowledgments

Part of this work was conducted while the authors were visiting the Simons Institute for the Theory of Computing. M.A. was supported by NSF awards CNS-2120667, CNS-2120603, CCF-1934846, and BU's Hariri Institute for Computing. This work was initiated while M.A. was affiliated with Boston University and Northeastern University and was done in part while M.A. was a research fellow at the Simons Institute for the Theory of Computing. P.I. was supported by the NSF TRIPODS program (award DMS-2022448) and Simons Investigator Award. R.R. was supported by the NSF TRIPODS program (award DMS-2022448) and CCF-2310818.

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

# A  Other related works

**General property testing of distributions:**    Testing properties of distributions has been extensively studied over the past few decades. Distribution testing under computational constraints has also been explored in [37, 70]. Hypothesis testing (and hypothesis selection) have received significant interest within the privacy community in machine learning [49, 21, 10, 4, 50, 72, 26, 27, 65]. Examples of other distributional properties that have been examined include testing monotonicity [20, 22, 11], testing histograms [24, 23], testing junta-ness [7, 32], and testing under structural properties [17, 58, 35, 44, 43].

**Connections to tolerant testing:**    Tolerant testing asks us *estimate* the true TV distance between a known distribution and another which we only have sample access to, up to a small additive error. Readers familiar with strong lower bounds in tolerant testing (see e.g., [78, 77, 25] where it's shown that $\Omega(n/\log(n))$ samples are needed in the case where the known distribution is uniform, compared to only $O(\sqrt{n})$ samples needed for uniformity testing) might find our results surprising. It is incorrect to conclude that our algorithm in Section B, i.e. the search component (which is adaptive to the distance between $\hat{p}$ and $p$), can perform tolerant testing between $p$ and $\hat{p}$. In reality, the algorithm finds an $\alpha$ which allows the testing component to terminate (either with accept or reject). Thus, while we know that the $\alpha$ found by our algorithm is never larger than $\|p - \hat{p}\|_{\mathrm{TV}}$, it could in fact be much lower than $\|p - \hat{p}\|_{\mathrm{TV}}$, meaning the $\alpha$ found by the search algorithm is not a good estimate for $\|p - \hat{p}\|_{\mathrm{TV}}$.

**Testable learning:**    Our framework bears some resemblance to *testable learning* as introduced in [71]. In this framework, the focus is on designing learning algorithms that can check whether the required underlying assumptions hold. If the assumptions do not hold, the algorithm may forego solving the problem. However, if it chooses to solve the problem, it must do so accurately, regardless of whether the assumptions hold. This is similar to our notion of testing with a suggested accuracy level, where the algorithm can either forego solving the problem if the assumption does not hold or solve it accurately regardless. Some examples of results in this framework are presented in [53, 54, 41, 55, 62, 61].

**Algorithms with predictions:**    Recently, there has been a burgeoning interesting in augmenting classical algorithm design with learned information. Most relevant to us are works which study learning-augmented algorithms under sublinear constraints, such as memory or sample complexity [57, 59, 60, 33, 47, 48, 31, 64, 73, 2]. We refer to the interested reader to the website `https://algorithms-with-predictions.github.io/` for an up-to-date collection of literature on learning-augmented algorithms.

# B  Searching for the appropriate accuracy level

Early on, we defined the augmented tester for testing identity, uniformity, and closeness. We generalize this concept for other properties of distributions. More formally, a property $\mathcal{P}$ is a set of distributions, and we say a distribution has property $\mathcal{P}$ iff it is in $\mathcal{P}$. The goal is to distinguish whether an unknown distribution $p$ is in $\mathcal{P}$, or it is far from all distributions that have the property. Similar to Definition 1.1, we define the notion of an augmented tester for $\mathcal{P}$ as follows:

**Definition B.1.** *Suppose we are given three parameters, $\alpha \in [0, 1]$, $\epsilon \in (0, 1)$, $\delta \in (0, 1)$. Assume $\mathcal{A}$ is an algorithm that receives all these parameters, and the description of a known distribution $\hat{p}$ as input, and it has sample access to an unknown underlying distribution $p$. We say algorithm $\mathcal{A}$ is an $(\alpha, \epsilon, \delta)$-augmented tester for property $\mathcal{P}$ for every $p$, $q$, and $\hat{p}$ over $[n]$:*

- *If $\hat{p}$ and $p$ are $\alpha$-close in total variation distance, the algorithm outputs* inaccurate *information with a probability at most $\delta/2$.*

- *If $p \in \mathcal{P}$, then the algorithm outputs* reject *with a probability at most $\delta/2$.*

- *If $p$ is $\epsilon$-far from every member of $\mathcal{P}$, then the algorithm outputs* accept *with a probability at most $\delta/2$.*

With this definition in mind, suppose we have an augmented tester for a property $\mathcal{P}$. Our goal is to find an appropriate $\alpha$ such that the augmented tester *solves* the problem: it outputs accept or reject, but also, it does not use too many samples. In this section, we introduce a search algorithm that seeks to find this appropriate $\alpha$. It is important to note that the $\alpha$ identified by our algorithm may *not* necessarily match the true accuracy level of the prediction, i.e., $\|p - \hat{p}\|_{\mathrm{TV}}$. Instead, it corresponds to the minimum number of samples that the augmented tester can solve the problem.

Our search algorithm runs in rounds, each set by a sample *budget* that increases as the process progresses. In round $i$, the algorithm determines an appropriate value for $\alpha_i$ and invokes the augmented tester, namely $\mathcal{A}$, with this chosen $\alpha_i$. The value of $\alpha_i$ is selected such that the augmented tester operates within the sample budget allocated for that round. If the tester outputs accept or reject, the search algorithm replicates this response. However, if the tester returns inaccurate information, we double the sample budget and proceed to the next round. The pseudocode for this procedure is provided in Algorithm 2. In the following theorem, we prove its performance.

---

**Algorithm 2** An augmented property tester without knowledge of $\alpha$

---

1: **procedure** AUGMENTED-TESTER-WITH-SEARCH($\epsilon, \delta'$, description of function $f$, algorithm $\mathcal{A}$)
2:      $s_{\min} \leftarrow \min_{\alpha \in [0,1]} f(\alpha, 1/3)$
3:      $s_{\max} \leftarrow \max_{\alpha \in [0,1]} f(\alpha, 1/3)$
4:      $t \leftarrow \left\lceil \log_2 \left( \frac{s_{\max}}{s_{\min}} \right) \right\rceil$
5:      $\delta \leftarrow \frac{\delta'}{t+1}$
6:      **for** $i = 0, 1, 2, \ldots, t - 1$ **do**
7:          $\alpha_i \leftarrow f^{-1} \left( 2^i \cdot s_{\min} \right)$
8:          $O \leftarrow$ the output of $\mathcal{A}$ with suggested accuracy level $\alpha_i$ and confidence parameter $\delta$
9:          **if** $O \in \{$accept, reject$\}$ **then**
10:              **return** $O$.
11:      Run the standard tester (without any prediction) with confidence parameter $\delta$ and return the answer.

---

**Theorem 6.** *Fix a parameter $\delta < 1/2$. Suppose $\mathcal{A}$ is an $(\alpha, \epsilon, \delta)$-augmented tester for property $\mathcal{P}$ that receives a suggested accuracy level $\alpha$ and confidence parameter $\delta$. Let $f : [0,1] \times [0,1] \to \mathbb{N}$ be a non-decreasing function for which $\mathcal{A}$ uses $f(\alpha, \delta)$[3] samples when it is invoked with parameter $\alpha$ and aims for the confidence parameter $\delta$. Algorithm 2 is a $(\epsilon, \delta')$-augmented tester, without knowledge of $\alpha$, for property $\mathcal{P}$ where*

$$\delta' := \delta \cdot \left( 1 + \left\lceil \log \left( \frac{\max_{\alpha \in [0,1]} f(\alpha)}{\min_{\alpha \in [0,1]} f(\alpha)} \right) \right\rceil \right)$$

*In addition, if $\alpha^*$ is the true accuracy level, then Algorithm 2 uses $O(f(\alpha^*))$ samples in expectation.*

*Proof.* First, we define the notation we use in this proof. Let $s_{\min}$ and $s_{\max}$ represent the smallest and largest sample sizes used by $\mathcal{A}$, respectively. $s_{\min}$ may be one or higher, and $s_{\max}$ is the sample size when the prediction did not make any improvements (the sample complexity of a standard tester). Let $t := \log_2 (s_{\max}/s_{\min})$. The algorithms runs in $t + 1$ rounds $i \in \{0, 1, \ldots, t\}$. In round $i < t$, our sample budget is $2^i \cdot s_{\min}$. We run the algorithm with a parameter $\alpha_i$ ensuring the sample complexity of $\mathcal{A}$ remains at most $2^{i-1} \cdot s_{\min}$ samples. In cases where multiple $\alpha$ values satisfy this criterion, we select the largest. We use (abuse in fact) the inverse function notation, defining $\alpha_i$ as $\alpha_i = f^{-1} \left( 2^i \cdot s_{\min} \right)$. If $\mathcal{A}$ finds an answer (accept or reject), we return the answer; Otherwise, we proceed to the next round. In round $i = t$, where we have $s_{\max} \approx 2^i \cdot s_{\min}$ samples, we run the standard tester and return its answer.

Now, we focus on proof of correctness. Note that we (may) call $\mathcal{A}$ $t$ times and the standard tester one time. Each of these tester works with probability at least $1 - \delta$. Thus, by the union bound, we can assume that they return the correct answer with probability at least $1 - \delta'$ where $\delta' := \delta/(t+1)$. As we have noted in Definition 1.1, $\mathcal{A}$ is resilient to inaccuracies, implying that even if the suggested accuracy level is not valid, if the tester does not output a false accept or reject with probability more than $\delta/2$. The same statement is correct for the standard tester. Now, our algorithm here replicates the

---

[3]Here, we omit the dependence to other non-varying parameters such as $\epsilon$ and $n$.

output that is produced by one of the testers. Thus, if they all of them outputs the correct answer, our algorithm outputs the correct answer as well. And, this event happens with probability at least $1 - \delta'$.

Next, we focus on the analysis of the sample complexity of this algorithm. Let $\alpha^*$ be the true prediction accuracy, and let $i^*$ be the first round where $\alpha_i \geq \alpha^*$. Recall that we assume that $\alpha_i$ is the largest $\alpha$ such that $f(\alpha_i) \leq 2^i \cdot s_{\min}$. In addition, we assume that $f$ is non-decreasing. Thus, we have[4]:

$$2^{i^*-1} \cdot s_{\min} < f(\alpha^*) \leq f(\alpha_{i^*}) \leq 2^{i^*} \cdot s_{\min}$$

If the algorithm ends at round $i$, we have used the following number of samples:

$$\text{\# samples for the first } i \text{ rounds} \leq \sum_{j=0}^{i} 2^j \cdot s_{\min} \leq 2^{i+1} \cdot s_{\min} \tag{4}$$
$$\leq 2^{i-i^*+2} \cdot \left( 2^{i^*-1} \cdot s_{\min} \right) \leq 2^{i-i^*+2} \cdot f(\alpha^*) .$$

On the other hand, the probability that algorithms end at round $i > i^*$ cannot be too high. If the algorithm ends at round $i$ or later, it means that $\mathcal{A}$ in rounds $i^*, i^*+1, \ldots, i-1$ must have returned inaccurate information. However, given our assumption, the true accuracy is not worse than $\alpha_i$, which makes outputting inaccurate information wrong. And, this event does not happen in each round with probability more than $\delta/2$ for each round independently. Let $I_{\text{end}}$ indicate a random variable that indicates the index of the round for which the algorithms ends. Thus, we have for every $i \geq i^*$:

$$\mathbf{Pr}[I_{\text{end}} \geq i] \leq \left( \frac{\delta}{2} \right)^{i-i^*} . \tag{5}$$

Now, we are ready to bound the expected value of the number of samples we use. Let $S$ denote the number of samples we use. Then, we have:

$$\mathbf{E}[S] = \sum_{i=0}^{t} \mathbf{E}[S \mid I_{\text{end}} = i] \cdot \mathbf{Pr}[I_{\text{end}} = i]$$
$$\leq 4 f(\alpha^*) \cdot \sum_{i=0}^{t} 2^{i-i^*} \cdot \mathbf{Pr}[I_{\text{end}} = i] \qquad \text{(by Eq. (4))}$$
$$\leq 4 f(\alpha^*) \cdot \left( \sum_{i=0}^{i^*} \mathbf{Pr}[I_{\text{end}} = i] + \sum_{i=i^*+1}^{t} \left( 1 + \sum_{j=0}^{i-i^*-1} 2^j \right) \cdot \mathbf{Pr}[I_{\text{end}} = i] \right)$$
$$= 4 f(\alpha^*) \cdot \left( \sum_{i=0}^{t} \mathbf{Pr}[I_{\text{end}} = i] + \sum_{i=i^*+1}^{t} \sum_{j=0}^{i-i^*-1} 2^j \cdot \mathbf{Pr}[I_{\text{end}} = i] \right)$$

---

[4]Without loss of generality assume, $f(\alpha_{-1}) = 0$.

Next, we write the second sum in terms of a new variable $k := i - i^* - 1$. Then, we swap the order of the summations on $k$ and $j$:

$$
\mathbf{E}[S] \leq 4\,f(\alpha^*) \cdot \left( 1 + \sum_{k=0}^{t-i^*-1} \sum_{j=0}^{k} 2^j \cdot \mathbf{Pr}[I_{\text{end}} = k + i^* + 1] \right)
$$

$$
\leq 4\,f(\alpha^*) \cdot \left( 1 + \sum_{j=0}^{t-i^*-1} 2^j \cdot \left( \sum_{k=j}^{t-i^*-1} \mathbf{Pr}[I_{\text{end}} = k + i^* + 1] \right) \right)
$$

$$
\leq 4\,f(\alpha^*) \cdot \left( 1 + \sum_{j=0}^{t-i^*-1} 2^j \cdot \mathbf{Pr}[I_{\text{end}} \geq j + i^* + 1] \right)
$$

$$
\leq 4\,f(\alpha^*) \cdot \left( 1 + \sum_{j=0}^{t-i^*-1} 2^j \cdot \left( \frac{\delta}{2} \right)^{j+1} \right) \qquad\qquad \text{(by Eq. (5))}
$$

$$
\leq 6 f(\alpha^*) . \qquad\qquad\qquad\qquad \text{(assuming } \delta < 1/2\text{)}
$$

$\square$

## C   Identity and uniformity testing

In this section, we focus on the problem of identity testing, which involves testing the equality between a known distribution $q$ and an unknown distribution $p$. Specifically, our goal is to determine whether $p = q$ or if they are $\epsilon$-far from each other. In an augmented setting, we are provided with another known distribution $\hat{p}$, predicted to represent $p$, along with a suggested level of accuracy $\alpha$. Surprisingly, our findings indicate that the sample complexity for this problem is influenced by a new parameter: the total variation distance between $q$ and $\hat{p}$, denoted by $d$. In particular, we have the following theorem:

**Theorem 7.** *Fix two arbitrary parameters $\epsilon \in (0, 1]$ and $\alpha \in [0, 1]$. Algorithm 3 is an $(\alpha, \epsilon, \delta = 0.1)$-augmented tester for identity that uses the following number of samples $s$, where:*

$$
s = \begin{cases} O\left( \frac{\sqrt{n}}{\epsilon^2} \right) & \text{if } d \leq \alpha \\[2mm] O\left( \min\left( \frac{1}{(d-\alpha)^2}, \frac{\sqrt{n}}{\epsilon^2} \right) \right) & \text{if } d > \alpha \end{cases} ,
$$

*and $d$ refers to $\|q - \hat{p}\|_{TV}$. For any $\alpha \in [0, 1]$, and $\epsilon \in [0, 1/2]$, we show the same number of samples is necessary for any $(\alpha, \epsilon, \delta = 2/3)$-augmented identity tester. In fact, the lower bound holds even when $q$ is a uniform distribution over $[n]$.*

The proof of this theorem follows from Proposition C.1, Proposition C.2, and Proposition C.3.

**Remark 8.** *A particular instance of this problem is* uniformity testing*, where $q$ is a uniform distribution over $[n]$. In Theorem 7, our upper bound applies to any arbitrary $q$. Furthermore, our tight lower bound is based on a hard instance, where $q$ is a uniform distribution. These results establish optimal upper and lower bounds for both identity testing and uniformity testing simultaneously.*

### C.1   Upper bound for identity testing

Our upper bound relies on a fundamental observation regarding the total variation distance: this distance is the maximum discrepancy between the probability masses that two distributions assign to any subset of their domain. Demonstrating a small total variation distance between two distributions entails proving that the discrepancy across every domain subset is minimal. In contrast, to prove a large total variation distance, one only needs to identify a single subset with a significant discrepancy. We employ the Scheffé set of $\hat{p}$ and $q$—defined as the set of elements $x \in [n]$ where $\hat{p}(x) < q(x)$, denoted by $S$—to serve as a witness for the farness of $p$ from either $q$ or $\hat{p}$. It is known that $S$ maximizing the discrepancy between the probability masses of $\hat{p}$ and $q$, implying:

$$
d := \|q - \hat{p}\|_{TV} = |q(S) - \hat{p}(S)| .
$$

In scenarios where the distance $d > \alpha$, accurately estimating the probability of $S$ according to $p$ with an accuracy of $(d - \alpha)/4$ provides a basis for distinguishing the farness from either $q$ or $\hat{p}$. If the probability masses assigned to $S$ by $q$ and $p$ differ by more than $(d - \alpha)/4$, it clearly indicates that $p$ is not identical to $q$, leading to outputting reject. Conversely, if the estimated probability of $p(S)$ is close to $q(S)$, then the discrepancy between $p(S)$ and $\hat{p}(S)$ must exceed $\alpha$, leading to outputting inaccurate information. The pseudocode of our algorithm presented in Algorithm 3. We prove the performance of our algorithm in the following proposition:

**Proposition C.1.** *Fix two arbitrary parameters* $\epsilon \in (0,1]$ *and* $\alpha \in [0,1]$. *Algorithm 3 is an* $(\alpha, \epsilon, \delta = 0.1)$-*augmented tester for identity that uses the following number of samples s, where:*

$$
s = \begin{cases} O\left(\frac{\sqrt{n}}{\epsilon^2}\right) & \text{if } d \le \alpha \\ O\left(\min\left(\frac{1}{(d-\alpha)^2}, \frac{\sqrt{n}}{\epsilon^2}\right)\right) & \text{if } d > \alpha \end{cases},
$$

*and* $d$ *refers to* $\|q - \hat{p}\|_{TV}$.

---

**Algorithm 3** An augmented tester for testing identity of $p$ and $q$ with a suggested value $\alpha$

---

1: **procedure** AUGMENTED-IDENTITY-TESTER($n$, $\alpha$, $\epsilon$, $\delta = 0.1$, $q$, $\hat{p}$, sample access to $p$)
2:      $d \leftarrow \|q - \hat{p}\|_{TV}$
3:      **if** $d \le \alpha$ or $\frac{1}{(d-\alpha)^2} > \frac{\sqrt{n}}{\epsilon^2}$ **then**
4:          **return** the standard testers answer.
5:      $S \leftarrow$ Scheffé set of $\hat{p}$ and $q$              $\triangleright x \in [n]$ is in the Scheffé set $S$ iff $\hat{p}(x) < q(x)$.
6:      Draw $m = O\left(\frac{1}{(d-\alpha)^2}\right)$ samples from $p$
7:      $\sigma \leftarrow$ fraction of samples that are in $S$
8:      **if** $|U_n(S) - \sigma| > \frac{d-\alpha}{4}$ **then**
9:          **return** reject.
10:     **else**
11:         **return** inaccurate information.

---

*Proof.* The proof of theorem is trivial in the setting where $d \le \alpha$ and $\frac{1}{(d-\alpha)^2} > \frac{\sqrt{n}}{\epsilon^2}$. In these cases, we use the standard tester for identity testing (e.g., [76, 30, 5]) with $\delta' = \delta/2 = 0.05$. Clearly, a wrong answer was produced with probability less than $\delta/2$. It is known that these tester only use $O(\sqrt{n}/\epsilon^2)$ samples.

Now, suppose $d > \alpha$. Simple application of Chernoff's bound shows that one can estimate the probability of the Scheffé set of $q$ and $\hat{p}$, $S$, up to error $(d - \alpha)/4$ with probability $1 - 0.95$ using $O(1/(d - \alpha)^2)$ samples. We refer to this estimate as $\sigma$, and we have with probability 0.95:

$$
|\sigma - p(S)| \le \frac{d - \alpha}{4} . \tag{6}
$$

Note that if $p = q$, then

$$
|\sigma - q(S)| = |\sigma - p(S)| \le \frac{d - \alpha}{4} .
$$

Hence, we do not output reject with probability more than $0.05 = \delta/2$. On the other hand, assume $\|\hat{p} - p\|_{TV} \le \alpha$, then we show it is unlikely for us to output inaccurate information.

$$
\begin{aligned}
\alpha \ge \|\hat{p} - p\|_{TV} &\ge |p(S) - \hat{p}(S)| \\
&\ge |\hat{p}(S) - q(S)| - |p(S) - q(S)| && \text{(by triangle inequality)} \\
&\ge \|q - \hat{p}\|_{TV} - |p(S) - q(S)| = d - |p(S) - q(S)| && \text{(by definition of Scheffé set)} \\
&\ge d - |\sigma - p(S)| - |\sigma - q(S)| \ge d - \frac{d - \alpha}{4} - |\sigma - q(S)| .
\end{aligned}
$$

                                                                  (by triangle inequality and Eq. (6))

Therefore, we obtain that with probability 0.95:

$$
|\sigma - q(S)| \ge \frac{3(d - \alpha)}{4} .
$$

Hence, we do not output inaccurate information with probability more than $0.05 = \delta/2$. Hence, Algorithm 3 is an $(\alpha, \epsilon, \delta = 0.1)$-agumented tester for identity. The sample complexity of the algorithm, in the case where $\alpha < d$ and $\frac{1}{(d-\alpha)^2} \leq \frac{\sqrt{n}}{\epsilon^2}$ is $O(1/(d-\alpha)^2)$. Thus, the proof is complete. $\square$

### C.2 Lower bound for uniformity testing when $\alpha \geq d$

In this section, we focus on the case where $\alpha \geq d$. We observed for this problem the number of samples depends on an additional parameter: the distance between $q$ and $\hat{p}$. If this distance is at most $\alpha$ then prediction does not help the algorithm. The algorithm would still be required to draw $\Omega(n/\epsilon^2)$ samples (which is the sufficient amount of sample for the standard case). Our proof is based on a reduction from standard identity testing to the augmented version of this problem which establishes the desired lower bound.

**Proposition C.2.** *Suppose we are given two known distributions $q$ and $\hat{p}$, and an unknown distribution $p$ over $[n]$. Let $d := \|q - \hat{p}\|_{TV}$. For any $\alpha$ in $[0,1]$, if $\alpha \geq d$, then any $(\alpha, \epsilon, \delta = 2/3)$-augmented tester requires $\Omega(\sqrt{n}/\epsilon^2)$ samples.*

*Proof.* We prove that standard identity testing can be reduced to augmented version of this problem if $\alpha \geq d$. For the standard tester, we are given a known distribution $q$ and an unknown distribution $p$. Let $\hat{p}$ be any arbitrary distribution that is in distance $d$ from $q$. Consider $\mathcal{A}$ as an $(\alpha, \epsilon, \delta = 2/3)$-augmented tester. The procedure for the standard tester is straightforward: Run $\mathcal{A}$ on $p$, $\hat{p}$, and $q$. If it returns accept, we also return accept; if it returns reject or inaccurate information, we return reject.

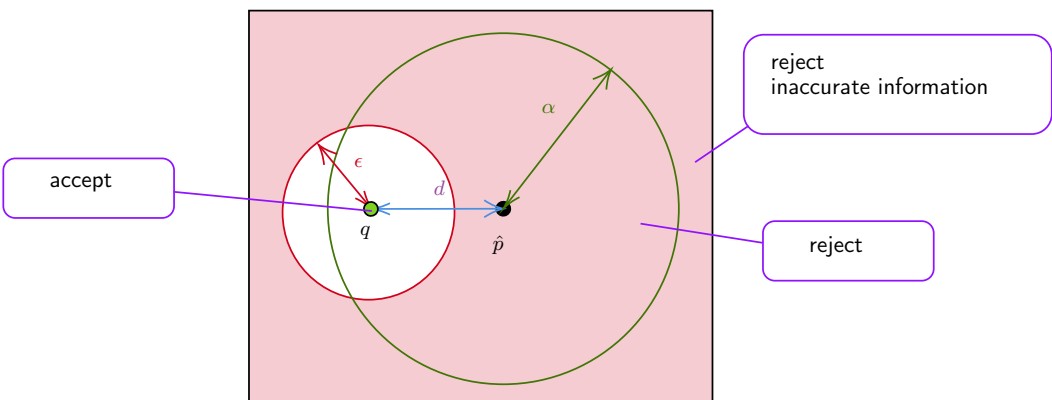

Figure 2: A diagram indicating the valid answer for the augmented tester $\mathcal{A}$ based on the total variation distances of $p$ from $q$ and $\hat{p}$ assuming $d \leq \alpha$. The standard tester requires to output accept if $p = q$, the green dot, and reject if $\|p - q\|_{TV} \geq \epsilon$, the red shaded region, with high probability. In addition, the augmented tester may output inaccurate information for when $\|p - q\|_{TV} \geq \epsilon$ and $\|p - \hat{p}\|_{TV} \geq \alpha$.

We show that the augmented tester distinguishes between the cases where $p = q$ and $\|p - q\|_{TV} > \epsilon$. See Figure 2. More precisely, Note that if $p = q$, then $p$ is within $\alpha$ distance of $\hat{p}$. Thus, the only valid answer of $\mathcal{A}$ in this case is accept. In this case, according to Definition 1.1, $\mathcal{A}$ returns reject and inaccurate information each with a probability of at most $\delta/2 = 1/6$. Therefore, by the union bound, we return accept with a probability of at least $2/3$. When $p$ and $q$ are $\epsilon$-far from each other, $\mathcal{A}$ returns accept with a probability of at most $\delta/2 = 1/6$. Consequently, we output the correct answer with a probability greater than $2/3$.

This reduction indicates that $\mathcal{A}$ must use at least as many samples as required for identity testing. Considering the existing lower bound for uniformity testing (where $q$ is a uniform distribution over $[n]$), augmented identity testing necessitates $O(\sqrt{n}/\epsilon^2)$ samples [69]. $\square$

## C.3 Lower bound for uniformity testing when $\alpha < d$

In this section, we consider the lower bound for the uniformity testing problem in the setting where $\alpha < d$. On the other hand, if $d - \alpha$ is not too small, the required lower bound is only $\Omega(1/(d-\alpha)^2)$. Otherwise, $\Omega(\sqrt{n}/\epsilon^2)$ samples is needed (as it is required for the standard tester).

At a high level, our proof consists of the following steps. First, we construct three distributions that *look* similar when we draw too few samples. We formalize the similarity of these three distributions using a multivariate coupling argument. Next, we describe valid answers for each of the distributions. The main message of this part is that there is no possible answer that is valid for all three distributions. Now, (similar to Le Cam's method), suppose we feed the algorithm with samples from one of these three distributions (each with probability 1/3). For any sample set, the algorithm outputs an answer (which may be randomized), however, this answer is considered as wrong for at least one of the underlying distributions. This is due to the fact that there is no universally valid answer that is simultaneously correct for all three distributions. Hence, if the algorithm outputs a valid answer with high probability, it must be able to distinguish the underlying distributions to some degree. On the other hand, the indistinguishably result says, it is impossible to tell these distributions apart. Thus, we reach a contradiction. And, the lower bound is concluded. More formally, we have the following proposition:

**Proposition C.3.** *Suppose we are given two known distributions $q$ and $\hat{p}$, and an unknown distribution $p$ over $[n]$. Let $d := \|q - \hat{p}\|_{TV}$. For any $\alpha \in [0, 1]$, $\epsilon \in [0, 1/2]$, and $d \in [0, 1/2]$, if $d > \alpha$ any $(\alpha, \epsilon, \delta = 2/3)$-augmented algorithm for testing identity of $p$ and $q$ with prediction $\hat{p}$ requires $s = \Omega\left(\min\left(\frac{1}{(d-\alpha)^2}, \frac{\sqrt{n}}{\epsilon^2}\right)\right)$ samples.*

*Proof.* Without loss of generality, assume $d - \alpha \leq 0.4$. Otherwise, we are required to establish a lower bound of $\Omega(1)$, which is necessary for any non-trivial testing problem.[5] As we have discussed it early, we prove this proposition in the following steps:

**Construction of distributions:** We assume $q = U_n$ is a uniform distribution over $[n]$. Without loss of generality assume $n$ is even. Otherwise, we can set the probability of one of the elements in all the distributions to zero. We define the prediction distribution as follows for every $i \in [n]$:

$$\hat{p}_i := \begin{cases} \dfrac{1 + 2d}{n} & i \text{ is even;} \\[2mm] \dfrac{1 - 2d}{n} & i \text{ is odd.} \end{cases}$$

It is not hard to assert that $\|\hat{p} - U_n\|_{TV}$ is $d$ satisfying the assumption we had in the statement of the proposition. Next, we construct two distributions $p^\bullet$ and $p^\diamond$.

Suppose we have a random vector $Z$ in $\{-1, +1\}^{n/2}$ where each coordinate $Z_i$ is one with probability 1/2 independently. Now, we define the following distributions over $[n]$:

$$p_i^\bullet := \begin{cases} \dfrac{1 + 2 Z_{i/2} \cdot \epsilon}{n} & i \text{ is even;} \\[2mm] \dfrac{1 - 2 Z_{(i+1)/2} \cdot \epsilon}{n} & i \text{ is odd.} \end{cases} \tag{7}$$

$$p_i^\diamond := \begin{cases} \dfrac{1 + 2(d - \alpha)}{n} & i \text{ is even;} \\[2mm] \dfrac{1 - 2(d - \alpha)}{n} & i \text{ is odd.} \end{cases} \tag{8}$$

It is not hard to see that the probabilities of each distributions sum up to one by observing the probability of two consecutive elements is $2/n$.

---

[5]As long as the problem does not have a pre-determined answer that works for all distributions, one must draw at least one sample.

**Indistinguishably of the distributions** Fix the number of samples $s$, and let $S_{|p}$ denote a random variable that is a sample set of size $s$ drawn from $p$. We show that $S_{|U_n}$, $S_{|p^\diamond}$, and $S_{|p^\bullet}$ are three random variables with small total variation distances between each pair. These distances are so small that it is practically impossible for the algorithm to tell them apart. We formalized this by providing a multivariate coupling between these random variables. We extend Le Cam's method for three random variables, and show that no algorithm with low error probability exists unless $s$ is large, establishing the desired lower bound for the problem.

**Lemma C.4.** *Suppose we are given the following parameters $\epsilon, d, \alpha, n$, and $s$. Assume $d - \alpha \leq 0.4$. Let $U_n$ denote the uniform distribution over $[n]$, and let $p^\bullet$ and $p^\diamond$ be the distributions defined in Equation (7) and Equation (8). Let $S_{|U_n}$, $S_{|p^\diamond}$, and $S_{|p^\bullet}$ be three random variables that are sample sets of size $s$ from each of these distributions. If $s \leq \min\left(\frac{0.00005}{(d-\alpha)^2}, \frac{0.004\sqrt{n}}{\epsilon^2}\right)$, then there exists a distribution over triples of three sample sets of size $s$ (that is $([n]^s)^3$), which we call a multivariate coupling $\mathcal{C}$, between $S_{|U_n}$, $S_{|p^\diamond}$, and $S_{|p^\bullet}$ such that:*

- *The marginals of $\mathcal{C}$ correspond to the three probability distributions $U_n^{\otimes s}$, $p^{\bullet \otimes s}$, and $p^{\diamond \otimes s}$. More precisely, for a sample $(S_1, S_2, S_3)$ drawn from $\mathcal{C}$, and every $S \in [n]^s$, we have:*

$$\mathbf{Pr}_{(S_1,S_2,S_3)\sim\mathcal{C}}[S_1 = S] = \mathbf{Pr}_{S_{|U_n}\sim U_n^{\otimes s}}\left[S_{|U_n} = S\right],$$

$$\mathbf{Pr}_{(S_1,S_2,S_3)\sim\mathcal{C}}[S_2 = S] = \mathbf{Pr}_{S_{|p^\bullet}\sim p^{\bullet \otimes s}}\left[S_{|p^\bullet} = S\right],$$

$$\mathbf{Pr}_{(S_1,S_2,S_3)\sim\mathcal{C}}[S_3 = S] = \mathbf{Pr}_{S_{|p^\diamond}\sim p^{\diamond \otimes s}}\left[S_{|p^\diamond} = S\right].$$

- $\mathbf{Pr}_{(S_1,S_2,S_3)\sim\mathcal{C}}[S_1 = S_2 = S_3] \geq 0.98.$

For the proof, see Section C.3.1. This lemma states that when the number of samples is too small, the sample sets looks very similar in all three cases: $p = U_n$, $p = p^\bullet$, and $p = p^\diamond$.

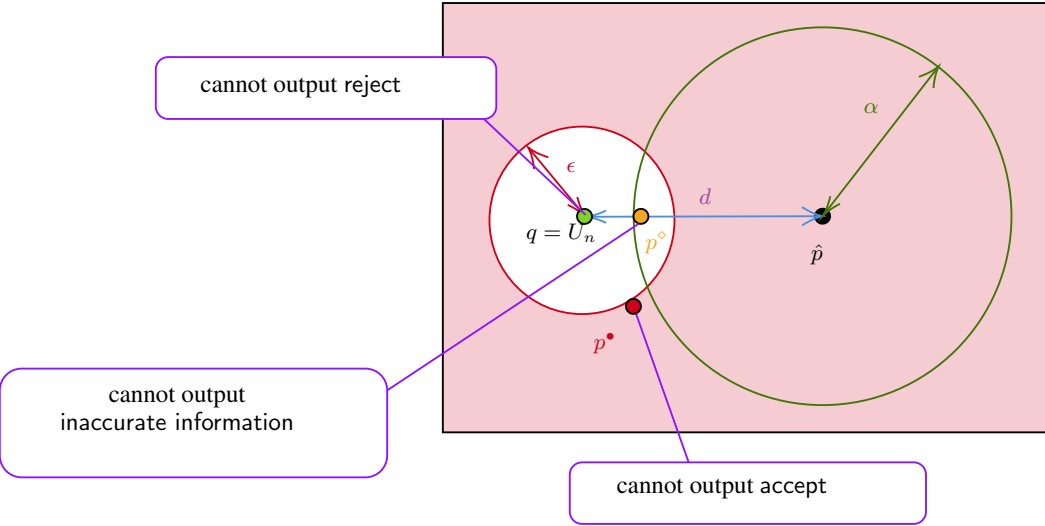

Figure 3: A diagram indicating the invalid answer for the three distributions $U_n$, $p^\bullet$, and $p^\diamond$.

**Valid answers for each distributions:** We focus on the valid output of the algorithm in these three cases using the Definition 1.1. According to this definition, we have:

- If $p = q$, then reject is not a valid answer.

- If $\|p - q\|_{\mathrm{TV}} > \epsilon$, then accept is not a valid answer.

- If $\|p - \hat{p}\|_{\mathrm{TV}} \leq \alpha$ then inaccurate information is not considered as a valid answer.

We denote the set of valid output for each case by $V_{|p}$. An accurate augmented tester with confidence parameter $\delta$, has to output an answer in $V_{|p}$ with probability at least $1 - \delta$.

1. $\boldsymbol{p = U_n}$ : In this case we have $p = q = U_n$. Clearly, in this case, reject is not a valid answer:
$$\text{reject} \notin V_{|U_n} .$$

2. $\boldsymbol{p = p^\bullet}$ : The total variation distance between $q = U_n$ and $p^\bullet$ is $\epsilon > \epsilon$. Thus, accept is not considered a valid output:
$$\text{accept} \notin V_{|p^\bullet} .$$

3. $\boldsymbol{p = p^\diamond}$ : It is not hard to see that $\|p^\diamond - \hat{p}\|_{\text{TV}} = \alpha$. Thus, inaccurate information is not a valid answer:
$$\text{inaccurate information} \notin V_{|p^\diamond} .$$

The diagram of the invalid outputs is shown in Figure 3 for an instance of these distributions. Let us explain what these valid sets are implying. The intersection of the three valid sets is empty, because among all possible outputs each set lacks at least one of them :
$$V_{|U_n} \cap V_{|p^\bullet} \cap V_{|p^\diamond} = \emptyset .$$

Therefore, there is no possible output that is considered valid on all the three distributions[6]. And, since these distributions are indistinguishable with high probability, no algorithm should perform well under all three possibilities.

**Deriving a contradiction.** Our proof is via contradiction. Suppose there exists an $(\alpha, \epsilon, \delta)$-augmented tester, called $\mathcal{A}$, for uniformity testing that uses $s$ samples.

$$3 \cdot (1 - \delta) \leq \mathbf{Pr}_{S_{|U_n} \sim U_n}\big[\mathcal{A}(S_{|U_n}) \in V_{|U_n}\big] + \mathbf{Pr}_{S_{|p^\bullet} \sim p^\bullet}\big[\mathcal{A}(S_{|p^\bullet}) \in V_{|p^\bullet}\big] + \mathbf{Pr}_{S_{|p^\diamond} \sim p^\diamond}\big[\mathcal{A}(S_{|p^\diamond}) \in V_{|p^\diamond}\big]$$

Let $I$ represent the internal coin tosses of the algorithm. Since the probability of $\mathcal{A}$ outputting an invalid answer is bounded by $\delta$, we have:
$$\delta \geq \mathbf{Pr}_{I, \, S_{|U_n} \sim U_n^{\otimes s}}\big[\mathcal{A}(S_{|U_n}) \notin V_{|U_n}\big] = \mathbf{E}_{S_{|U_n} \sim U_n^{\otimes s}}\big[\mathbf{Pr}_I\big[\mathcal{A}(S_{|U_n}) \notin V_{|U_n}\big]\big]$$
$$= \mathbf{E}_{S_{|U_n} \sim U_n^{\otimes s}}\big[\mathbf{Pr}_I\big[\mathcal{A}(S_{|U_n}) = \text{reject}\big]\big] = \mathbf{E}_{(S_1, S_2, S_3) \sim \mathcal{C}}\big[\mathbf{Pr}_I[\mathcal{A}(S_1) = \text{reject}]\big]$$
$$\geq \mathbf{E}_{(S_1, S_2, S_3) \sim \mathcal{C}}\big[\mathbf{Pr}_I[\mathcal{A}(S_1) = \text{reject}] \mid S_1 = S_2 = S_3\big] \cdot \mathbf{Pr}_{(S_1, S_2, S_3) \sim \mathcal{C}}\big[S_1 = S_2 = S_3\big] .$$
$$\text{(by law of total expectation)}$$

We can write the above equation for $p^\bullet$ and $p^\diamond$ and sum of them up. Thus, we obtain:

$$\delta \geq \frac{1}{3} \cdot \mathbf{Pr}_{(S_1, S_2, S_3) \sim \mathcal{C}}[S_1 = S_2 = S_3] \cdot \mathbf{E}_{(S_1, S_2, S_3) \sim \mathcal{C}}\big[\mathbf{Pr}_I[\mathcal{A}(S_1) = \text{reject}] + \mathbf{Pr}_I[\mathcal{A}(S_2) = \text{accept}]$$
$$+ \mathbf{Pr}_I[\mathcal{A}(S_3) = \text{ inaccurate information}] \mid S_1 = S_2 = S_3\big]$$
$$= \frac{1}{3} \cdot \mathbf{Pr}_{(S_1, S_2, S_3) \sim \mathcal{C}}[S_1 = S_2 = S_3] .$$

For the last line above, note that when $S_1 = S_2 = S_3$, the terms inside the expectations are the sum of probabilities of all possible outputs for $\mathcal{A}$. Hence, the sum of these probabilities is one.

Recall that earlier, we assumed that $\delta = 0.3$, and using the properties of the coupling, we know that $S_1 = S_2 = S_3$ with probability at least 0.98. Thus, we get:
$$0.3 = \delta \geq \frac{1}{3} \cdot \mathbf{Pr}_{(S_1, S_2, S_3) \sim \mathcal{C}}[S_1 = S_2 = S_3] \geq \frac{0.98}{3} > 0.32 .$$

Hence, we reach a contradiction. Thus, no such algorithm like $\mathcal{A}$ exists. □

---

[6]It is worth noting that a pair of valid sets may have some overlap. That is, the algorithm can output an answer that is correct for both of the distributions. In other words, technically the algorithm does not have to distinguish any pair of these distributions as long as there is a valid answer which works for both of them.

### C.3.1 Multivariate coupling between the three hard distributions

In this section, we prove the lemma on coupling that we used to prove our lower bound. This lemma implies indistinguishability between $S_{|U_n}$, $S_{|p^\bullet}$, and $S_{|p^\diamond}$. For a definition and some basics on coupling, see Section F.3.

**Lemma C.4.** *Suppose we are given the following parameters $\epsilon, d, \alpha, n$, and $s$. Assume $d - \alpha \leq 0.4$. Let $U_n$ denote the uniform distribution over $[n]$, and let $p^\bullet$ and $p^\diamond$ be the distributions defined in Equation (7) and Equation (8). Let $S_{|U_n}$, $S_{|p^\diamond}$, and $S_{|p^\bullet}$ be three random variables that are sample sets of size $s$ from each of these distributions. If $s \leq \min\left(\frac{0.00005}{(d-\alpha)^2}, \frac{0.004\sqrt{n}}{\epsilon^2}\right)$, then there exists a distribution over triples of three sample sets of size $s$ (that is $([n]^s)^3$), which we call a multivariate coupling $\mathcal{C}$, between $S_{|U_n}$, $S_{|p^\diamond}$, and $S_{|p^\bullet}$ such that:*

- *The marginals of $\mathcal{C}$ correspond to the three probability distributions $U_n^{\otimes s}$, $p^{\bullet \otimes s}$, and $p^{\diamond \otimes s}$. More precisely, for a sample $(S_1, S_2, S_3)$ drawn from $\mathcal{C}$, and every $S \in [n]^s$, we have:*

$$\mathbf{Pr}_{(S_1,S_2,S_3)\sim\mathcal{C}}[S_1 = S] = \mathbf{Pr}_{S_{|U_n}\sim U_n^{\otimes s}}\left[S_{|U_n} = S\right],$$

$$\mathbf{Pr}_{(S_1,S_2,S_3)\sim\mathcal{C}}[S_2 = S] = \mathbf{Pr}_{S_{|p^\bullet}\sim p^{\bullet \otimes s}}\left[S_{|p^\bullet} = S\right],$$

$$\mathbf{Pr}_{(S_1,S_2,S_3)\sim\mathcal{C}}[S_3 = S] = \mathbf{Pr}_{S_{|p^\diamond}\sim p^{\diamond \otimes s}}\left[S_{|p^\diamond} = S\right].$$

- $\mathbf{Pr}_{(S_1,S_2,S_3)\sim\mathcal{C}}[S_1 = S_2 = S_3] \geq 0.98$.

*Proof.* We start by bounding the total variation distance between two pairs of distributions: $\|U_n^{\otimes s} - p^{\bullet \otimes s}\|_{\text{TV}}$, and $\|U_n^{\otimes s} - p^{\diamond \otimes s}\|_{\text{TV}}$.

In [69], the author has shown that the total variation distance between $S_{|U_n}$ and $S_{|p^\bullet}$ is bounded by:

$$\|U_n^{\otimes s} - p^{\bullet \otimes s}\|_{\text{TV}} \leq \frac{\sqrt{\exp\left(\frac{s^2 \cdot (2\epsilon)^4}{n}\right) - 1}}{2} \leq 0.01. \tag{9}$$

The last inequality above comes from our assumption that $s \leq \frac{0.004\sqrt{n}}{\epsilon^2}$.

Next, we focus on the total variation distance between $S_{|p^\diamond}$ and $S_{|U_n}$. Note that $p^\diamond$ is a distribution that has some bias towards the even elements, but it is uniform on both sets of odd and even elements. Therefore, to bound the total variation distance between $U_n^{\otimes s}$ and $p^{\diamond \otimes s}$, we use a similar argument for showing a lower bound of $\Omega(1/\Delta)^2$ to distinguish whether a coin is fair or it has a bias of $(1 + \Delta)/2$. Below is our formal argument.

It is not hard to see that one can use the following process to generate a sample from $p^\diamond$. First, we draw a sample $X \sim \mathbf{Ber}(1/2 + (d - \alpha))$ from the Bernoulli distribution with a success probability $1/2 + (d - \alpha)$. If $X = 1$, we pick an even element (uniformly); Otherwise, we pick an odd element. Similarly, we can draw samples from a uniform distribution as follows: draw $X' \sim \mathbf{Ber}(1/2)$ from the Bernoulli distribution with a success probability $1/2$. If $X' = 1$, we pick an even element (uniformly); Otherwise, we pick an odd element. One can view this process of generating a sample in $[n]$ from a binary variable ($X$ or $X'$) as a *channel*. The data processing inequality says that the distance (more precisely any $f$-divergences) between two random variables does not increases after they pass through a channel. Hence, we use this fact and bound the total variation distance between $p^{\diamond \otimes s}$ and $U_n^{\otimes s}$:

$$\|U_n^{\otimes s} - p^{\diamond \otimes s}\|_{\text{TV}} \leq \sqrt{\frac{1}{2} \cdot KL\left(U_n^{\otimes s} \| p^{\diamond \otimes s}\right)} \qquad \text{(by Pinsker's inequality)}$$

$$\leq \sqrt{\frac{s}{2} \cdot KL\left(U_n \| p^\diamond\right)} \qquad \text{(since samples are drawn i.i.d.)}$$

$$\leq \sqrt{\frac{s}{2} \cdot KL\left(\mathbf{Ber}(1/2) \| \mathbf{Ber}(1/2 + (d - \alpha))\right)}.$$

$$\text{(by data processing inequality)}$$

$$\leq \sqrt{2s} \cdot (d - \alpha) \qquad \text{(when } 2(d - \alpha) \leq 0.8)$$

Since we had this assumption where $s$ is at most $\frac{0.00005}{(d-\alpha)^2}$, we get.

$$\|U_n^{\otimes s} - p^{\diamond \otimes s}\|_{\text{TV}} \leq \sqrt{2\,s} \cdot (d - \alpha) \leq 0.01 \tag{10}$$

So far, we have shown that $\|U_n^{\otimes s} - p^{\bullet \otimes s}\|_{\text{TV}}$ and $\|U_n^{\otimes s} - p^{\diamond \otimes s}\|_{\text{TV}}$ are at most 0.01. Using the coupling lemma (See Fact F.3), there exist two maximal couplings $\mathcal{C}^\bullet$ and $\mathcal{C}^\diamond$ between the following pairs of distributions: $(U_n^{\otimes s}, p^{\bullet \otimes s})$ and $(U_n^{\otimes s}, p^{\diamond \otimes s})$. The properties of these maximal couplings are:

- The marginals of the couplings are equal to the two pairs of probability distributions. That is:

$$\mathcal{C}_1^\bullet = U_n^{\otimes s}, \text{ and } \mathcal{C}_2^\bullet = p^{\bullet \otimes s},$$
$$\mathcal{C}_1^\diamond = U_n^{\otimes s}, \text{ and } \mathcal{C}_2^\diamond = p^{\diamond \otimes s}.$$

- $\mathbf{Pr}_{(S_1, S_2) \sim \mathcal{C}^\bullet}[S_1 = S_2] \geq 0.99$, and $\mathbf{Pr}_{(S_1, S_2) \sim \mathcal{C}^\diamond}[S_1 = S_2] \geq 0.99$.

For every $S \in [n]^m$, let $\mathcal{C}_{2|S}^\diamond$ be a probability distribution over $[n]^m$. The probability of $S'$ according to $\mathcal{C}_{2|S}^\diamond$ is:

$$\mathcal{C}_{2|S}^\diamond(S') := \mathbf{Pr}_{(S_1, S_2) \sim \mathcal{C}^\diamond}[S_2 = S'|S_1 = S].$$

With these definitions in mind, we create our multivariate coupling $\mathcal{C}$ by essentially joining the two couplings $\mathcal{C}^\bullet$ and $\mathcal{C}^\diamond$. The following randomized procedure produces a random sample $(S_1, S_2, S_3)$ from $\mathcal{C}$:

1. Let $(S_1, S_2)$ be a sample from $\mathcal{C}^\bullet$.

2. Let $S_3$ be a sample from $\mathcal{C}_{2|S_1}^\diamond$.

3. Output $(S_1, S_2, S_3)$ as a sample from $\mathcal{C}$.

Given the definition of $\mathcal{C}^\bullet$, it is clear that the first two marginals of $\mathcal{C}$ are exactly $U_n^{\otimes s}$ and $p^{\bullet \otimes s}$. To see that the third marginal of $\mathcal{C}$ is equal to $p^\diamond$, we have the following for every $S' \in [n]^m$:

$$\mathbf{Pr}[S_3 = S'] = \sum_{S_1 \in [n]^m} \mathbf{Pr}_{(S_1', S_2') \sim \mathcal{C}^\bullet}[S_1' = S_1] \cdot \mathcal{C}_{2|S_1}^\diamond(S')$$

$$= \sum_{S_1 \in [n]^m} \mathbf{Pr}_{(S_1', S_2') \sim \mathcal{C}^\bullet}[S_1' = S_1] \cdot \mathbf{Pr}_{(S_1'', S_2'') \sim \mathcal{C}^\diamond}[S_2'' = S'|S_1'' = S_1]$$

$$= \sum_{S_1 \in [n]^m} \mathbf{Pr}_{(S_1', S_2') \sim \mathcal{C}^\diamond}[S_1' = S_1] \cdot \mathbf{Pr}_{(S_1'', S_2'') \sim \mathcal{C}^\diamond}[S_2'' = S'|S_1'' = S_1]$$

$$= \sum_{S_1 \in [n]^m} \mathcal{C}^\diamond(S_1, S') = \mathcal{C}_2^\diamond = p^\diamond(S').$$

Now, we show that $S_1$, $S_2$, and $S_3$ are equal with high probability. This stems from the fact that $\mathcal{C}^\bullet$ and $\mathcal{C}^\diamond$ were maximal couplings, and with high probability, the pairs drawn from them are equal. More formally, via the union bound, we have:

$$\mathbf{Pr}_{(S_1, S_2, S_3) \sim \mathcal{C}}[S_1, S_2, S_3] \geq 1 - \mathbf{Pr}_{(S_1, S_2, S_3) \sim \mathcal{C}}[S_1 \neq S_2] - \mathbf{Pr}_{(S_1, S_2, S_3) \sim \mathcal{C}}[S_1 \neq S_3]$$

$$= \mathbf{Pr}_{(S_1, S_2, S_3) \sim \mathcal{C}}[S_1 = S_3] - \mathbf{Pr}_{(S_1, S_2) \sim \mathcal{C}^\bullet}[S_1 \neq S_2]$$

$$= \sum_{S \in [n]^m} \mathbf{Pr}_{(S_1, S_2, S_3) \sim \mathcal{C}}[S_1 = S \text{ and } S_3 = S] - \mathbf{Pr}_{(S_1, S_2) \sim \mathcal{C}^\bullet}[S_1 \neq S_2]$$

$$= \sum_{S \in [n]^m} \mathcal{C}^\diamond(S, S) - \mathbf{Pr}_{(S_1, S_2) \sim \mathcal{C}^\bullet}[S_1 \neq S_2]$$

$$\text{(With a similar argument as above)}$$

$$= \mathbf{Pr}_{(S_1, S_2) \sim \mathcal{C}^\diamond}[S_1 = S_2] - \mathbf{Pr}_{(S_1, S_2) \sim \mathcal{C}^\bullet}[S_1 \neq S_2]$$

$$\geq 0.99 - (1 - 0.99) = 0.98. \qquad \text{(by Eq. 9 and Eq. 10)}$$

Hence, the proof is complete.

$\square$

# D  Augmented closeness testing

In this section, we focus on the problem of closeness testing in the augmented setting. More formally, we have the following theorem.

**Theorem 9.** *For every $\alpha$ and $\epsilon$ in $(0, 1)$, there exists an algorithm for augmented closeness testing that uses $\Theta\left(\frac{n^{2/3}\alpha^{1/3}}{\epsilon^{4/3}} + \frac{\sqrt{n}}{\epsilon^2}\right)$ samples and succeeds with probability at least $2/3$. In addition, any algorithm for this task is required to use $\Omega\left(\frac{n^{2/3}\alpha^{1/3}}{\epsilon^{4/3}} + \frac{\sqrt{n}}{\epsilon^2}\right)$ samples.*

For the proof of the upper bound see section 3. And, for the proof of lower bound see section D.1.

## D.1  Lower bound for closeness testing

In this section, we prove a lower bound for augmented closeness testing where a known distribution $\hat{p}$ is provided satisfying $\|p - \hat{p}\|_{\mathrm{TV}} = \alpha$. We have the following theorem.

**Theorem 10.** *Suppose we are given $\alpha$ and $\epsilon$ in $(0, 1)$. Any $(\alpha, \epsilon, \delta := 11/24)$-augmented tester for closeness testing of distributions over $[n]$ uses $\Omega\left(\frac{n^{2/3}\alpha^{1/3}}{\epsilon^{4/3}} + \frac{\sqrt{n}}{\epsilon^2}\right)$ samples. The lower bound holds even when $\alpha$ is provided to the algorithm.*

*Proof.* The $\Omega(\sqrt{n}/\epsilon^2)$ term comes from the existing lower bound for uniformity testing since we can reduce uniformity testing to augmented closeness testing as follows: Suppose we wish to test whether a distribution $q$ over $[n]$ is uniform or $\epsilon$-far from uniform. Suppose $\mathcal{A}$ is an $(\alpha, \epsilon, 11/24)$-augmented closeness tester. Set $p$ to be the uniform distribution over $[n]$. Let $\hat{p}$ be any distribution that is $\alpha$-far from uniform. For instance, $\hat{p}$ can be a distribution that is $(1 + \alpha)/n$ on half of the elements and $(1 - \alpha)/n$ on the other half. Since $\mathcal{A}$ is an augmented tester, it can be used to distinguish whether $p = q$ or $\|p - q\|_{\mathrm{TV}} > \epsilon$ with probability $11/24$.

Upon receiving $\hat{p}$ and samples from $p$ and $q$, if $\mathcal{A}$ returns inaccurate information or reject, we output reject; otherwise, we output accept. Given Definition 1.1, if $q$ is uniform, we will not output inaccurate information or reject with probability more than $11/24$. And, if $q$ is $\epsilon$-far from the uniform distribution (equivalently $p$), we will not output accept with probability more than $11/48$. Hence, we have a tester with confidence parameter $11/48$ for uniformity testing. Thus, $s$ must be $\Omega(\sqrt{n}/\epsilon^2)$, which is the number of samples required for uniformity testing [69].

To prove the lower bound of $\Omega\left(n^{2/3}\alpha^{1/3}/\epsilon^{4/3}\right)$ samples, we only need to focus on the case where $\alpha \geq (\sqrt{n}\epsilon^2)^{-1}$. Otherwise, $n^{2/3}\alpha^{1/3}/\epsilon^{4/3} = O(\sqrt{n}/\epsilon^2)$, making the lower bound trivial due to the $\Omega(\sqrt{n}/\epsilon^2)$ lower bound shown earlier. We consider the following cases depending on $\epsilon$:

Case 1: $\epsilon \leq (\sqrt{n}\epsilon^2)^{-1}$. Therefore, in this case, the interesting regime of parameter for $\alpha$ is that $\alpha \geq \epsilon$. We prove the lower bound for this case in Theorem 11 in Section D.1.1.

Case 2: $\epsilon > (\sqrt{n}\epsilon^2)^{-1}$. In this case, in addition to considering the case where $\alpha \geq \epsilon$, we also need to study the case where $\alpha$ is in $\left((\sqrt{n}\epsilon^2)^{-1}, \epsilon\right)$. We prove the lower bound for this case in Theorem 12 in Section D.1.2.

This completes the proof. $\square$

### D.1.1  Lower bound for $\alpha \geq \epsilon$

Below, we provide a lower bound for augmented closeness testing when $\alpha \geq \epsilon$. At a high level, Theorem 11 achieves this lower bound by creating a challenging instance. This involves embedding the difficult instances of the standard closeness testing problem into another distribution, spanning a domain of $[n-1]$ elements, while concentrating the majority of the distribution's mass on the last

element. The prediction, being a singleton distribution centered on the last element, does not facilitate solving the standard closeness testing problem, which is embedded within the first $[n-1]$ elements.

**Theorem 11.** *Suppose we are given $\alpha$ and $\epsilon$ in $(0,1)$. Assume we have two unknown distributions $p$ and $q$ and a known distribution $\hat{p}$ over the domain $[n]$. If $\alpha \geq \epsilon$, then any $(\alpha, \epsilon, \delta := 11/24)$- augmented tester for testing closeness of $p$ and $q$ requires the following number of samples:*

$$s = \Omega\left(\frac{n^{2/3}\,\alpha^{1/3}}{\epsilon^{4/3}} + \frac{\sqrt{n}\,\alpha}{\epsilon^2}\right).$$

*Proof.* Our proof is based on a reduction from standard closeness testing to augmented closeness testing. Assume there exists an algorithm, $\mathcal{A}$, that upon receiving $\hat{p}$ and $s$ samples from $p$ and $q$, can distinguish whether $p = q$ or $\|p - q\|_{\mathrm{TV}} \geq \epsilon$ with probability 11/24. We show that this algorithm can then be used to test closeness of two distributions, $\tilde{p}$ and $\tilde{q}$ over $[n-1]$ with proximity parameter $\epsilon' := \epsilon/\alpha$.

Suppose we have sample access to $\tilde{p}$ and $\tilde{q}$. We construct $p$ and $q$ based on $\tilde{p}$ and $\tilde{q}$ as follows:

$$p_i = \alpha \cdot \tilde{p}_i \quad \forall i \in [n-1]\,, \qquad p_n = 1 - \alpha\,,$$

$$q_i = \alpha \cdot \tilde{q}_i \quad \forall i \in [n-1]\,, \qquad q_n = 1 - \alpha\,.$$

Note that given a sample from $\tilde{p}$ (respectively $\tilde{q}$), we can generate a sample from $p$ (respectively $q$) by outputting that sample with probability $\alpha$ and $n$ otherwise. Also, if $\tilde{p}$ is equal to $\tilde{q}$, then $p$ and $q$ are equal. And, if $\|\tilde{p} - \tilde{q}\|_{\mathrm{TV}} \geq \epsilon'$, then $\|p - q\|_{\mathrm{TV}} \geq \epsilon$ due to the following:

$$\|p - q\|_{\mathrm{TV}} = \frac{1}{2} \sum_{i \in [n-1]} |p_i - q_i| = \frac{\alpha}{2} \sum_{i \in [n-1]} |\tilde{p}_i - \tilde{q}_i| = \alpha \cdot \|\tilde{p} - \tilde{q}\|_{\mathrm{TV}}.$$

Let $\hat{p}$ be the singleton distribution on $n$ (i.e., $\hat{p}_n = 1$). In this case, $\|p - \hat{p}\|_{\mathrm{TV}}$ equals $\alpha$. We feed $\mathcal{A}$ with $\hat{p}$ and samples from $p$ and $q$. Now, if $\mathcal{A}$ outputs $p = q$, we declare $\tilde{p} = \tilde{q}$. Otherwise, we declare $\tilde{p}$ is $\epsilon'$-far from $\tilde{q}$. Given our construction, this answer is true with probability 11/24.

Note that in this process, in expectation, we draw $\alpha s$ samples from $\tilde{p}$ and $\tilde{q}$. Thus, using Markov's inequality, the probability of drawing more than $100\,\alpha\,s$ samples from each of $\tilde{p}$ and $\tilde{q}$ is at most 0.01. Hence, using $\mathcal{A}$, there exists an algorithm for testing closeness of $\tilde{p}$ and $\tilde{q}$ with probability $1 - (11/24 + 0.01) > 0.53$ that uses $100\,\alpha\,s$ samples from $\tilde{p}$ and $\tilde{q}$. Using the existing lower bound for the closeness testing problem (see Theorem 4 in [69] and Proposition 4.1 in [30]), we have:

$$100\,\alpha\,s = \Omega\left(\frac{n^{2/3}}{\epsilon'^{4/3}} + \frac{\sqrt{n}}{\epsilon'^2}\right) = \Omega\left(\frac{n^{2/3}\,\alpha^{4/3}}{\epsilon^{4/3}} + \frac{\sqrt{n}\,\alpha^2}{\epsilon^2}\right).$$

Thus, by dividing by $100\alpha$, we get the following bound as desired:

$$s = \Omega\left(\frac{n^{2/3}\,\alpha^{1/3}}{\epsilon^{4/3}} + \frac{\sqrt{n}\,\alpha}{\epsilon^2}\right).$$

□

### D.1.2   Lower bound for $\alpha$ in $\left(c \cdot (\sqrt{n}\epsilon^2)^{-1}, \epsilon\right)$

We extend the existing lower bound for closeness testing to the augmented setting. In the standard setting, the lower bound for closeness testing is derived from the hard instances for uniformity testing with one key adjustment: introducing elements with large probability ($\approx n^{-2/3}$) in the distributions. These large elements share identical probability masses in both $p$ and $q$, indicating that they do not contribute to the distance between the two distributions. However, their presence in the sample set *confuses* the algorithm: Because of their large probabilities, their behaviour in the sample set may be deceivingly hint "not uniform", making it difficult for the algorithm to decide whether the rest of the distributions are uniform or not. Therefore, the algorithm needs $s \approx n^{2/3}$ samples to first identify these large elements, and then it can test the uniformity on the rest of the distributions. The surprising part about this result is that $s \approx n^{2/3}$ is much larger than the $\sqrt{n}$ samples which suffice for testing uniformity.

The challenge in our case is that $\hat{p}$ may give away the large elements to the algorithm. Hence, to prove the lower bound, we create hard instances by adding as many large elements as possible, without changing $\hat{p}$ by limiting the overall probability mass of the large elements to $\alpha$.

Here is our instance: we assume $\hat{p}$ is the uniform distribution; $p$ assigns $\approx (1 - \alpha)/n$ probability mass to $O(n)$ elements chosen at random, and assigns $\approx n^{-2/3}$ probability mass to $\alpha \cdot n^{2/3}$ many elements in the domain. Now, $q$ has two options. Half the time, $q$ is equal to $p$. The other half, $q$ shares the same large elements, but it is not quite uniform on the rest of the distributions. In particular, $q$ assigns probabilities $(1 \pm \Theta(\epsilon))(1 - \alpha)/n$. to the chosen $O(n)$ elements randomly, making it $\epsilon$-far from $p$.[7] Using this construction, we use a result of Valiant [78] to show that these two cases are indistinguishable unless we draw $\Omega(n^{2/3}\alpha^{1/3})$ samples. More precisely, we have the following theorem:

**Theorem 12.** *Suppose we are given the following parameters: a positive integer $n$, $\epsilon \in (0, 1/6)$, and $\alpha \in \big((\sqrt{n}\epsilon^2)^{-1}, \epsilon\big)$ (with this implicit assumption that $(\sqrt{n}\epsilon^2)^{-1}$ is less than $\epsilon$). There exists a distribution $\hat{p}$ and a family of pairs of distribution $p$ and $q$ such that any augmented algorithm for closeness testing must use*

$$\Omega\left(\frac{n^{2/3}\,\alpha^{1/3}}{\epsilon^{4/3}}\right)$$

*samples.*

*Proof.* Let $k$ be an integer that is at most

$$k \leq c_k \left(\frac{n^{2/3}\,\alpha^{1/3}}{\epsilon^{4/3}}\right) \tag{11}$$

for a sufficiently small absolute constant $c_k$ which we determine later. Assume we have $k$ samples from $p$ and $q$ that are available to the algorithm. We show that there are two (families of) distributions that any augmented tester has to distinguish them with high probability. However, they are information-theoretically indistinguishable using only $k$ samples.

**Setting up parameters:** First, we focus on determining a series of parameters which we use in our proof. We define $\ell$ as follows:

$$\ell := \left\lfloor \frac{n^{2/3}\,\alpha^{4/3}}{2\,\epsilon^{4/3}} \right\rfloor - \mathbb{1}_{n \neq \left\lfloor \frac{n^{2/3}\,\alpha^{4/3}}{2\,\epsilon^{4/3}} \right\rfloor (\bmod\, 2)}. \tag{12}$$

It is not hard to see that $\ell$ is an integer. For our proof, we require that $\ell$ and $n$ are either both even or both odd. Here, the role of the indicator variable is to reduce $\ell$ by one in case they are not. Also, $\ell$ is positive, since due to the range of $\alpha$ and $\epsilon$, we have:

$$\frac{n^{2/3}\,\alpha^{4/3}}{2\,\epsilon^{4/3}} > \frac{n^{2/3}}{2\,\epsilon^{4/3}} \cdot \left(\frac{1}{\sqrt{n}\,\epsilon^2}\right)^{4/3} = \frac{1}{2\,\epsilon^4} \geq \frac{6^4}{2}.$$

Given this definition, there exists an absolute constant $c_\ell < 0.5$, such that $\ell = c_\ell \cdot n^{2/3}\,\alpha^{4/3}/\epsilon^{4/3}$. We use this identity as the definition of $\ell$ throughout this proof. Given the assumption about the range of $\alpha$, we also have:

$$\ell = \frac{c_\ell \cdot n^{2/3}\alpha^{4/3}}{\epsilon^{4/3}} < c_\ell\, n^{2/3} \leq \frac{n}{2}. \tag{13}$$

Let $L$ denote a set of $\ell$ random elements in $[n]$ to be our large elements. $\overline{L} := [n] \setminus L$ denotes the set of small elements. Since $n$ and $\ell$ are either both even or both odd, $\overline{L}$ has even number of elements. Now, partition $\overline{L}$ to two random subset of the same size $\overline{L}_1$ and $\overline{L}_2$. Set $\epsilon' := 6\,\epsilon < 1$. Set

$$c_k := \min\left(\frac{1}{5080320 \cdot c_\ell},\ \frac{c_\ell}{2000},\ \frac{1}{2000}\right).$$

---

[7]Our parameters in this discussion are not precise. See our proof for a rigorous setting of parameters.

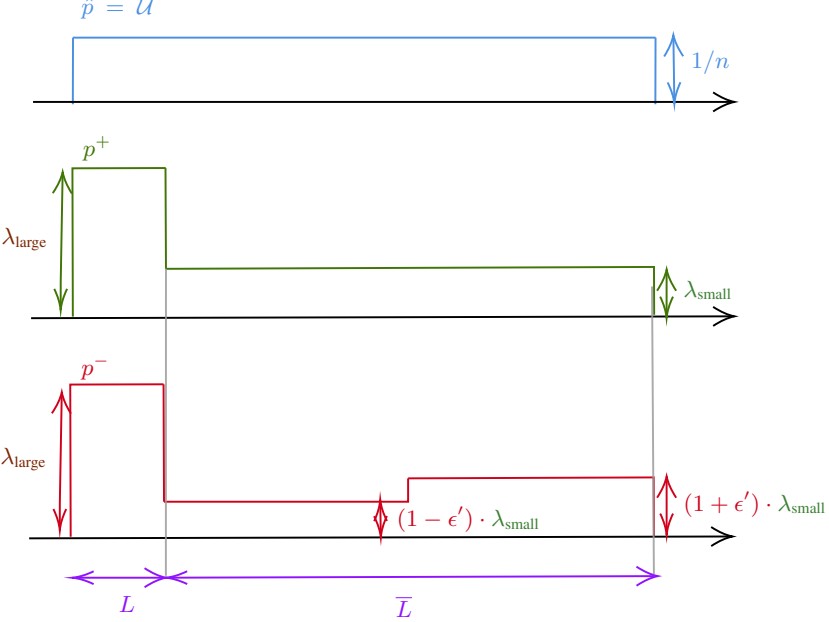

Figure 4: A visualization of $\hat{p}$, $p^+$, and $p^-$.

**Constructing the distributions:** We set $\hat{p}$ to be a uniform distribution over $[n]$. That is, $\hat{p}_i = 1/n$ for every $i \in [n]$. We construct two distributions $p^+$ and $p^-$ as follows. For every $i \in [n]$, the probability mass of $i$ according to these distributions are as follows:

$$p_i^+ := \begin{cases} \dfrac{1}{n} + \dfrac{\alpha}{\ell} & i \in L \\ \dfrac{1}{n} - \dfrac{\alpha}{n-\ell} & i \in \overline{L} \end{cases} \qquad p_i^- := \begin{cases} \dfrac{1}{n} + \dfrac{\alpha}{\ell} & i \in L \\ (1+\epsilon') \cdot \left( \dfrac{1}{n} - \dfrac{\alpha}{n-\ell} \right) & i \in \overline{L}_1 \\ (1-\epsilon') \cdot \left( \dfrac{1}{n} - \dfrac{\alpha}{n-\ell} \right) & i \in \overline{L}_2 \end{cases}$$

For a visual representation of these distributions, refer to Figure 4. It is not hard to verify that the probability masses sum up to one since $|L| = \ell$, and $|\overline{L}| = n - \ell$. Also, these probabilities are non-negative due to Equation (13) and the fact that $\alpha < \epsilon \leq 1/6$:

$$\frac{1}{n} - \frac{\alpha}{n-\ell} \geq \frac{1}{n} - \frac{\alpha}{n/2} > 0.$$

Also, given these definitions, $p^+$ is $\alpha$-close to $\hat{p}$, and $p^+$ and $p^-$ are $\epsilon$-far from each other:

$$\|p^+ - \hat{p}\|_{\mathrm{TV}} = \frac{1}{2} \left( \sum_{i \in L} \frac{\alpha}{\ell} + \sum_{i \in \overline{L}} \frac{\alpha}{n-\ell} \right) = \alpha,$$

$$\|p^+ - p^-\|_{\mathrm{TV}} = \frac{1}{2} \sum_{i \in \overline{L}} \epsilon' \cdot \left( \frac{1}{n} - \frac{\alpha}{n-\ell} \right) = \frac{\epsilon'}{2} \cdot \left( \frac{n-\ell}{n} - \alpha \right) > \frac{\epsilon'}{2} \cdot \left( \frac{1}{2} - \frac{1}{6} \right) \geq \epsilon.$$

**Proof of indistinguishability:** Now, consider two pairs of distributions $(p = p^+, q = p^+)$ and $(p = p^+, q = p^-)$. Given the distances above any algorithm for augmented closeness testing should output accept on $(p^+, p^+)$, and reject on $(p^+, p^-)$. To establish our lower bound, we show these pairs of distributions are indistinguishable with high probability. We use the moment based indistinguishably result for symmetric properties[8] of pairs of distributions in [78].

---

[8]A symmetric property of a pair of distribution is a property that does not change if we permute the domain elements of two distribution via the same permutation. The closeness testing of $p$ and $q$ in our context is a

Before stating their theorem, we need to define the notion of the $(k, k)$-based moments of a pair distributions $(p, q)$ as follows:

**Definition D.1** ($(k, k)$-based moments). *For every positive integer $k$ the $(k, k)$-based moments of a pair of distributions $p$ and $q$ are the following values:*

$$M_{a,b}^{(k)}(p, q) := k^{a+b} \sum_{i \in [n]} p_i^a q_i^b$$

*for every non-negative integers $a$ and $b$.*

**Fact D.2** (Adapted from [78]). *Given an integer $k$, suppose we have four distributions, $p^{(1)}, p^{(2)}, q^{(1)}$, and $q^{(2)}$ over $[n]$ where the maximum probability that each of them assigns to an element is less than $1/(1000\,k)$. If*

$$\sum_{a+b \geq 2} \frac{\left| M_{a,b}^{(k)}\left(p^{(1)}, q^{(1)}\right) - M_{a,b}^{(k)}\left(p^{(2)}, q^{(2)}\right) \right|}{\lfloor \frac{a}{2} \rfloor! \, \lfloor \frac{b}{2} \rfloor! \, \sqrt{1 + \max\left( M_{a,b}^{(k)}\left(p^{(1)}, q^{(1)}\right), \, M_{a,b}^{(k)}\left(p^{(2)}, q^{(2)}\right) \right)}} < \frac{1}{360}, \qquad (14)$$

*then no algorithm can distinguish the pair $\left(p^{(1)}, q^{(1)}\right)$ from $\left(p^{(2)}, q^{(2)}\right)$ with a success probability more than $13/24$ upon receiving $\mathrm{Poi}(k)$ samples from each distribution in the pair.*

First, we verify that all the domain elements of $p^+$ and $p^-$ have probability masses at most $1/(1000\,k)$ according to both $p^+$ and $p^-$. Recall that $\epsilon < 1$. Observe that it suffices to show that $1/n < 1/(2000\,k)$ and $\alpha/\ell < 1/(2000\,k)$. The first inequality holds due to the fact that $\alpha$ is in $\left( (\sqrt{n}\epsilon^2)^{-1}, \epsilon \right)$:

$$\frac{k}{n} = \frac{c_k\, n^{2/3}\, \alpha^{1/3}}{n\, \epsilon^{4/3}} = c_k \cdot \left( \frac{\alpha}{(\sqrt{n}\epsilon^2)^2} \right)^{1/3} < c_k \cdot \left( \alpha\, \epsilon^2 \right)^{1/3} < c_k \cdot \epsilon < c_k \leq \frac{1}{2000}$$

And, the second inequality comes from the definitions of $\ell$, $k$, and $c_k$:

$$\frac{\alpha}{\ell} = \frac{\alpha\, \epsilon^{4/3}}{c_\ell\, n^{2/3}\, \alpha^{4/3}} = \frac{\epsilon^{4/3}}{c_\ell\, n^{2/3}\, \alpha^{1/3}} = \frac{c_k}{c_\ell\, k} \leq \frac{1}{2000\,k}.$$

For the rest of this proof, we focus on showing Equation (14) for the pairs $(p^+, p^+)$ and $(p^+, p^-)$. For simplicity in our equations, we use the following notation:

$$\lambda_{\text{large}} := \frac{1}{n} + \frac{\alpha}{\ell}, \qquad \lambda_{\text{small}} := \frac{1}{n} - \frac{\alpha}{n - \ell}. \qquad (15)$$

Using the Definition D.1, it is not hard to see that:

$$M_{a,b}^{(k)}\left(p^+, p^+\right) = k^{a+b} \left( \sum_{i \in L} \lambda_{\text{large}}^{a+b} + \sum_{i \in \overline{L}} \lambda_{\text{small}}^{a+b} \right) = k^{a+b} \left( \ell \cdot \lambda_{\text{large}}^{a+b} + (n - \ell) \cdot \lambda_{\text{small}}^{a+b} \right)$$

$$M_{a,b}^{(k)}\left(p^+, p^-\right) = k^{a+b} \left( \sum_{i \in L} \lambda_{\text{large}}^{a+b} + \sum_{i \in \overline{L}_1} (1 + \epsilon')^b \cdot \lambda_{\text{small}}^{a+b} + \sum_{i \in \overline{L}_2} (1 - \epsilon')^b \cdot \lambda_{\text{small}}^{a+b} \right)$$

$$= k^{a+b} \left( \ell \cdot \lambda_{\text{large}}^{a+b} + (n - \ell) \cdot \lambda_{\text{small}}^{a+b} \cdot \left( \frac{(1 + \epsilon')^b + (1 - \epsilon')^b}{2} \right) \right).$$

Therefore, we get:

$$\left| M_{a,b}^{(k)}\left(p^+, p^+\right) - M_{a,b}^{(k)}\left(p^+, p^-\right) \right| = (n - \ell) \cdot \lambda_{\text{small}}^{a+b} \cdot \left( \frac{(1 + \epsilon')^b + (1 - \epsilon')^b}{2} - 1 \right).$$

---

symmetric property since permuting the labels of the elements does not change the distance between $p$ and $q$; Furthermore, $\hat{p}$ is identical for every permutation of domain elements.

On the other hand, we have:

$$\sqrt{1 + \max\left( M_{a,b}^{(k)}\left(p^{(1)}, q^{(1)}\right), \ M_{a,b}^{(k)}\left(p^{(2)}, q^{(2)}\right)\right)} \geq \sqrt{k^{a+b} \cdot \ell \cdot \lambda_{\text{large}}{}^{a+b}}.$$

Using the two equations above, we can bound the right hand side of Equation (14):

$$\text{RHS of Eq. (14)} \leq \frac{n-\ell}{\sqrt{\ell}} \sum_{a+b \geq 2} \frac{k^{(a+b)/2} \cdot \lambda_{\text{small}}{}^{a+b}}{\lfloor \frac{a}{2} \rfloor! \ \lfloor \frac{b}{2} \rfloor! \ \lambda_{\text{large}}{}^{(a+b)/2}} \cdot \left( \frac{(1+\epsilon')^b + (1-\epsilon')^b}{2} - 1 \right)$$

Note that for $b = 0$ or $1$, the term in the sum is zero. Hence, we only need to iterate over all $b \geq 2$ and $a \geq 0$. Let $T$ denote $\sqrt{k/\lambda_{\text{large}}} \cdot \lambda_{\text{small}}$. Then, we can write the above equation in the following form:

$$\text{RHS of Eq. (14)} \leq \frac{n-\ell}{\sqrt{\ell}} \cdot \left( \sum_{a \geq 0} \frac{T^a}{\lfloor \frac{a}{2} \rfloor!} \right) \cdot \left( \frac{1}{2} \sum_{b \geq 2} \frac{(T \cdot (1+\epsilon'))^b}{\lfloor \frac{b}{2} \rfloor!} + \frac{1}{2} \sum_{b \geq 2} \frac{(T \cdot (1-\epsilon'))^b}{\lfloor \frac{b}{2} \rfloor!} - \sum_{b \geq 2} \frac{(T)^b}{\lfloor \frac{b}{2} \rfloor!} \right)$$

To bound these terms, we have the following lemma:

**Lemma D.3.** *For $x \in \mathbb{R}$, we have*

$$\sum_{i=0}^{\infty} \frac{x^i}{\lfloor \frac{i}{2} \rfloor!} = (1+x)e^{x^2},$$

*and for $x \in [0, 1.33]$, we have:*

$$x^2 + x^3 \leq \sum_{i=2}^{\infty} \frac{x^i}{\lfloor \frac{i}{2} \rfloor!} < x^2 + x^3 + 3\,x^4.$$

*Proof.* We start by splitting the terms corresponding to odd and even $i$'s in the sum, and obtain the following:

$$\sum_{i=0}^{\infty} \frac{x^i}{\lfloor \frac{i}{2} \rfloor!} = \sum_{j=0}^{\infty} \frac{x^{2j}}{j!} + \frac{x^{2j+1}}{j!} = (1+x) \sum_{j=0}^{\infty} \frac{x^{2j}}{j!} = (1+x) \cdot e^{x^2}.$$

The last equality above is due to the Taylor expansion of $e^y = 1 + y/1! + y^2/2! + \dots$ about $y = 0$. Next, we have:

$$\sum_{i=2}^{\infty} \frac{x^i}{\lfloor \frac{i}{2} \rfloor!} = \sum_{i=0}^{\infty} \frac{x^i}{\lfloor \frac{i}{2} \rfloor!} - x - 1 = (1+x) \cdot \left( e^{x^2} - 1 \right).$$

Now, if $0 < y \leq 1.79$, then the following inequality holds: $y < e^y - 1 < y + y^2$. Thus, for $x \in (0, 1.33)$, we have:

$$x^2 + x^3 = (1+x) \cdot x^2 \leq (1+x) \cdot \left( e^{x^2} - 1 \right) < (1+x) \cdot (x^2 + x^4) = x^2 + x^3 + (1+x)\,x^4 < x^2 + x^3 + 3\,x^4.$$

$\square$

Now, assume $T \cdot (1 + \epsilon') \leq 1.33$. We continue bounding the right hand side of Equation (14) via the above fact:

$$\text{RHS of Eq. (14)} \leq \frac{n-\ell}{\sqrt{\ell}} \cdot \left( \sum_{a \geq 0} \frac{T^a}{\lfloor \frac{a}{2} \rfloor !} \right) \cdot \left( \frac{1}{2} \sum_{b \geq 2} \frac{(T \cdot (1+\epsilon'))^b}{\lfloor \frac{b}{2} \rfloor !} + \frac{1}{2} \sum_{b \geq 2} \frac{(T \cdot (1-\epsilon'))^b}{\lfloor \frac{b}{2} \rfloor !} - \sum_{b \geq 2} \frac{(T)^b}{\lfloor \frac{b}{2} \rfloor !} \right)$$

$$< \frac{n}{\sqrt{\ell}} \cdot \left( (1+T) \cdot e^{T^2} \right) \cdot \left( \frac{1}{2} \left( T^2 \cdot (1+\epsilon')^2 + T^3 \cdot (1+\epsilon')^3 + 3\,T^4 \cdot (1+\epsilon')^4 \right) \right.$$

$$+ \frac{1}{2} \left( T^2 \cdot (1-\epsilon')^2 + T^3 \cdot (1-\epsilon')^3 + 3\,T^4 \cdot (1-\epsilon')^4 \right) - T^2 - T^3 \Big)$$

$$= \frac{n}{\sqrt{\ell}} \cdot 14 \cdot \left( \left( \frac{(1+\epsilon')^2 + (1-\epsilon')^2}{2} - 1 \right) \cdot T^2 + \left( \frac{(1+\epsilon')^3 + (1-\epsilon')^3}{2} - 1 \right) \cdot T^3 \right.$$

$$+ \left( \frac{3\,(1+\epsilon')^4 + 3\,(1-\epsilon')^4}{2} \right) \cdot T^4 \Big)$$

$$\leq \frac{n}{\sqrt{\ell}} \cdot 14 \cdot \left( \epsilon'^2\,T^2 + 3\epsilon'^2 \cdot T^3 + 24\,T^4 \right) \ . \qquad\qquad \text{(using } \epsilon' \leq 1\text{)}$$

We can bound $T$ from above given the trivial bounds we have that $\lambda_{\text{small}} \leq 1/n$ and $\lambda_{\text{large}} \geq \alpha/\ell$ by their definitions in Equation (15):

$$T := \sqrt{\frac{k}{\lambda_{\text{large}}}} \cdot \lambda_{\text{small}} \leq \sqrt{\frac{k \cdot \ell}{\alpha}} \cdot \frac{1}{n} \leq \frac{\sqrt{c_\ell \cdot c_k} \cdot (n^{2/3}\alpha^{1/3}/\epsilon^{4/3})}{n} = \sqrt{c_\ell \cdot c_k} \cdot \left( \frac{\alpha}{n\,\epsilon^4} \right)^{1/3} \quad (16)$$

$$\leq \sqrt{c_\ell \cdot c_k} \cdot \alpha < 6\,\epsilon = \epsilon' \ .$$

The second to last inequality above is due to the range we assumed for $\alpha$: it is greater than $(\sqrt{n}\epsilon^2)^{-1}$. For the last inequality, we assumed that $\alpha < \epsilon$. Moreover, using $c_k \leq (5080320 \cdot c_\ell)^{-1}$, we have $\sqrt{c_\ell \cdot c_k} < 6$. Therefore, we obtain:

$$\text{RHS of Eq. (14)} < \frac{n}{\sqrt{\ell}} \cdot 14 \cdot 28 \cdot (\epsilon'\,T)^2 \qquad\qquad \text{(using } 1 \geq \epsilon' > T \text{ in Eq. (16))}$$

$$\leq \left( 14 \cdot 28 \cdot 6^2 \right) \cdot \frac{n}{\sqrt{\ell}} \cdot \epsilon^2 \cdot \frac{k \cdot \ell}{\alpha} \cdot \frac{1}{n^2}$$

$$\text{(since } \epsilon' := 6\,\epsilon \text{ and } T \leq \sqrt{k \cdot \ell/\alpha}/n \text{ in middle of Eq. (16))}$$

$$= 14112 \cdot \frac{\epsilon^2 k}{\alpha\,n} \cdot \sqrt{\ell} \leq (14112 \cdot c_\ell) \cdot \frac{\epsilon^{4/3}\,k}{n^{2/3}\,\alpha^{1/3}} \quad \text{(using } \ell := c_\ell\,n^{2/3}\alpha^{4/3}/\epsilon^{4/3})$$

$$\leq 14112 \cdot c_\ell \cdot c_k \leq \frac{1}{360} \ .$$

For the last inequality, recall that early on we set $k$ to be at most $c_k \left( n^{2/3}\,\alpha^{1/3}/\epsilon^{4/3} \right)$ and $c_k$ is defined to be at most $(5080320 \cdot c_\ell)^{-1}$. Therefore, the requirements in Fact D.2 hold as desired in our setting. Thus, the proof is complete. $\qquad\square$

# E   Empirical Evaluation

We evaluate our algorithm empirically on real and synthetic data on the closeness testing problem. Recall that given samples to two unknown distributions $p$ and $q$ over the domain $[n]$, we wish to test if $p = q$ or if $\|p - q\|_{\text{TV}} \geq \epsilon$. They are referred to as the $(p, p)$ case or the $(p, q)$ case respectively. Our experiments test the performance of our algorithm and baselines.

**Datasets.**

- **Hard Instance**: This is a simplified version of the hard instance described in Section D.1 for Theorem 12. A conceptually similar hard instance is also used in the classical closeness testing setting without any learned augmentation, as detailed in Section D.1.2
  Let $[n]$ be the domain. In the instance used in the experiments, $p$ and $q$ agree on the first $m := \lfloor n^{2/3} \rfloor$ elements. Both distributions have probability mass $1/(2m)$ on these domain elements. $p$

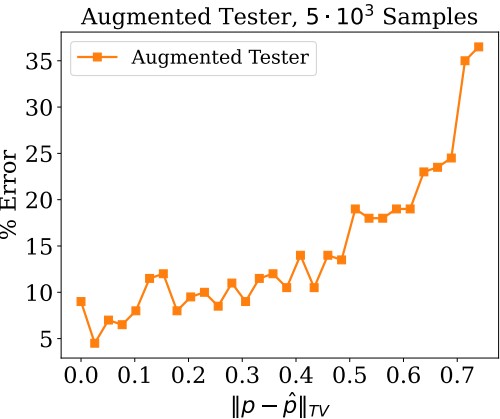

Figure 5: Error as a function of prediction quality for the 'Hard Instance' dataset

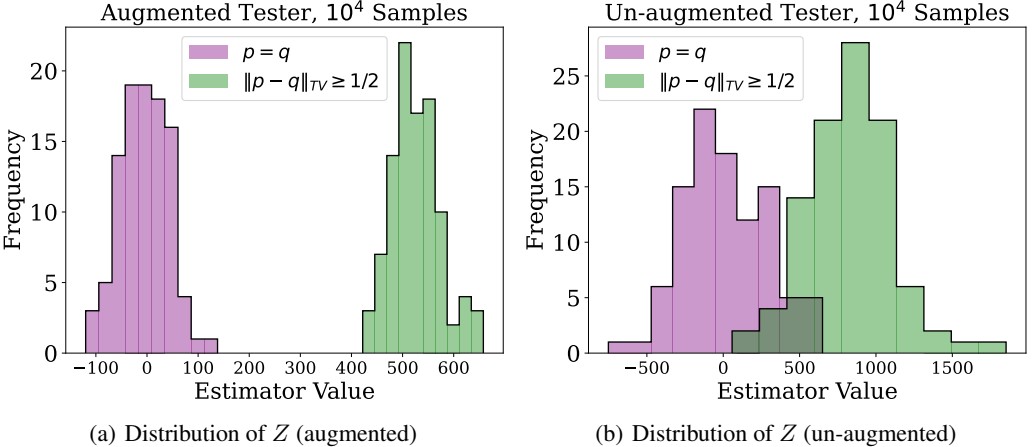

(a) Distribution of $Z$ (augmented)

(b) Distribution of $Z$ (un-augmented)

Figure 6: Additional figures for the 'Hard Instance' dataset

also has probability mass $2/n$ on domain elements between $n/2$ and $3n/4$ and $q$ has probability mass $2/n$ on domain elements between $3n/4$ and $n$. All other probability masses are $0$. We can check that $\|p-q\|_{\mathrm{TV}} = 1/2$. The first $m$ domain elements represent 'heavy hitters' which conspire to fool any testing algorithm and the rest of the domain are items which contribute to all the TV distance but are rarely sampled. Indeed, we built upon this intuition to prove a formal lower bound in Theorem 12.

We consider hints of the form $\hat{p} = (1-\beta) \cdot p + \beta \cdot \mathrm{Unif}(n)$, where $\mathrm{Unif}(n)$ is the uniform distribution on $[n]$ elements and $\alpha \in [0, 1]$ is an interpolation parameter. $\alpha = 0$ corresponds to a perfect hint and the quality of the hint degrades as $\alpha$ increase. When $\alpha = 1$, we receive $\hat{p} = \mathrm{Unif}(n)$, i.e. a hint which not require any learned information to instantiate. In our experiments, we set $n$, the domain size, to $n = 10^6$ and our goal is to determine if we are sampling from $(p, p)$ or $(p, q)$.

- **IP**: The data is internet traffic data collected at a backbone link of a Tier1 ISP in a data center[9]. Using the preprocessing in [1], we construct the empirical distribution over source IP addresses in consecutive chunks of time. Each distribution represents approximately $170ms$ of empirical network traffic data. The support size of each distribution is $\approx 45,000$ while the total domain size is $> 2.5 \cdot 10^6$. Distributions curated from network traffic closer in time are closer in TV distance. This dataset has been used in other distribution testing literature [1, 48].

---

[9]From CAIDA internet traces 2019, see `https://www.caida.org/catalog/datasets/monitors/passive-equinix-nyc`

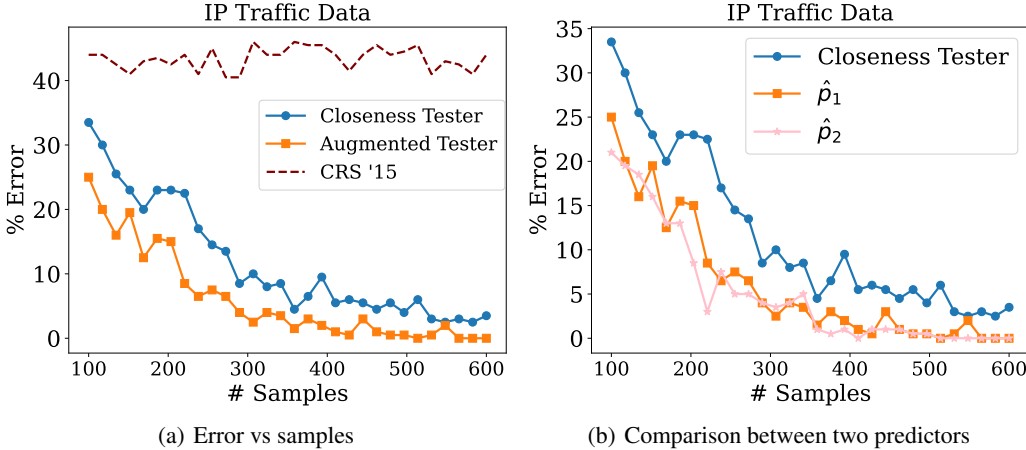

(a) Error vs samples          (b) Comparison between two predictors

Figure 7: Results for the IP dataset

We set $p$ to be distribution for the first chunk of time and we let $q$ be the distribution for the last chunk of time. Empirically, we have $\|p - q\|_{\text{TV}} \approx 0.72$. The hint is for $p$, denoted as $\hat{p}$, is the second chunk of time. Unlike the previous case, the hint is relatively far from $p$ in terms of TV distance ($\|p - \hat{p}\|_{\text{TV}} \approx 0.59$). Nevertheless, they share similar 'heavy' items (IP addresses with large probability mass), since $p$ and $\hat{p}$ represent distributions of network traffic which are close in time.

**Baselines.** We compare our main algorithm with the following baselines.

- **CRS '15**. This is the state of the art algorithm with access to *almost-perfect* predictions from [28]. More precisely, they assume predictions to *both* distributions $p$ and $q$, denoted as $\hat{p}$ and $\hat{q}$. Furthermore, they require $\hat{p}(i) = (1 \pm \varepsilon/100)p(i)$ and $\hat{q}(i) = (1 \pm \varepsilon/100)q(i)$ for all $i \in [n]$. Note that we only assume access to (much weaker) predictions for only $p$.

- **Closeness Tester**. This is the state of the art algorithm of [42] without any predictor access. We refer to this algorithm as the standard or the un-augmented tester as well.

**Error Measurement.** At a high level, both our algorithm (denoted as 'Augmented Tester' and the prior SOTA un-augmented approach, (abbreviated as 'Closeness Tester') compute an estimator $Z$ based on the samples requested. Then, both algorithms threshold the value of $Z$ to determine which case we are in (either $p = q$ or $\|p - q\|_{\text{TV}} \geq \varepsilon$. Note that $Z$ is a random variable in both cases (since it depends on random samples). Ideally, the distribution of $Z$ is *well-separated* in the two different hypotheses.

Thus in our experiments, given a sample budget, we compute the empirical distribution of $Z$ for both algorithms under both hypothesis cases across 100 trials. For a fixed algorithm, our error measurement computes the dissimilarity of the two empirical histograms: if the histograms are 'well-separated' then we know the algorithm can adequately distinguish the two hypotheses. On the other hand, if the histograms of $Z$ in the two cases are highly overlapping, then it is clear that the algorithm cannot distinguish well. Thus, given the two empirical histograms, we calculate the best empirical threshold which best separates them. This is done for both algorithms, ours and the classical Closeness Tester. Then we measure the fraction of data points in the histogram which are misclassified as the error. Calculating the best empirical threshold also makes both of the algorithms more practical as the theoretical thresholds are only stated asymptotically with un-optimized constant factors, making the theoretical values impractical. Finally, note that it is trivial to get $50\%$ error.

CRS '15 is a fundamentally different algorithm. Rather than calculating an estimator, it queries the hints $\hat{p}$ and $\hat{q}$ on the samples received and check if they are consistent with each other. If all samples $i$ satisfy $\hat{p}(i) = (1 \pm \varepsilon/10)\hat{q}(i)$, the algorithm declares we are in the $p = q$ case.

Note that as implemented, our algorithm and the standard Closeness Tester do not require the knowledge of $\varepsilon$ ($\varepsilon$ is not needed to calculate the estimator $Z$). On the other hand CRS '15 crucially

requires the knowledge of $\varepsilon$. Thus, we make this baseline even stronger by providing it with the exact knowledge of $\varepsilon$ (which the other two algorithms do not have access to).

**Results.**

- **Hard Instance**. Our results are shown in Figures 1 and 5. We now describe the plots. Figure 1 displays the performance of all algorithms as a function of the number of samples seen. Note that the $y$ axis measures the fraction of times we can distinguish the two hypotheses across 100 trials. In Figure 1, our algorithm has access to $\hat{p} = p$ and CRS '15 has access to $\hat{p} = p$. Furthermore, it has access to a hint $\hat{q}$ which satisfies $\|\hat{q} - q\|_{TV} < 0.002$. $\hat{q}$ hat is created by removing a small fraction of the mass on a small fraction of the heavy domain elements between 1 and $m$.

  We see that the performance of both our Augmented Tester and the standard Closeness Tester improves as the sample complexity increases. On the other hand, CRS '15 essentially always obtains the trivial guarantee of $50\%$ error even when it is fed a large number of samples. This is because with decent probability, we sample an element where $\hat{q}(i) = 0$ and $\hat{p}(i) \neq 0$. Thus in the $p = q$ case, it is very likely to output the incorrect answer.

  Figure 1 also demonstrates that our augmented tester achieves up to $> \mathbf{20x}$ reduction in error than the standard Closeness Tester. It also requires $> \mathbf{2x}$ fewer samples to achieve $0\%$ error empirically.

  Our improvements are due to the fact that the empirical estimators of $Z$ in the two cases of $p = q$ and $p \neq q$ are very well-separated for the Augmented Tester. In fact, as shown in Figures 6(a) and 6(b), if we fix the number of samples to $10^4$, the empirical distributions of $Z$ in the two cases do not overlap at all for the augmented algorithm. On the other hand, thee is a significant overlap for the un-augmented tester (the standard Closeness Tester). This implies that the Closeness Tester baseline cannot distinguish the two cases as well as the augmented algorithm.

- **IP Data**. Our results are shown in Figure 7(a). Again we see the same qualitative behavior: CRS '15 is quite brittle to prediction errors: in the case $(p, p)$ case, we feed CRS '15 with the perfect prediction $p$ for the first distribution, and $\hat{p}$ for the second. In this case, it typically outputted the incorrect answer. Furthermore, our augmented algorithm has a better sample vs error trade-off than the un-augmented approach. Indeed, we observed that while the augmented tester only required $\sim 500$ samples to obtain $0\%$ error in 100 trials, the standard tester required $> 700$ samples, implying our augmented approach is $> \mathbf{40\%}$ more efficient.

**Robustness to Prediction Error.** Our experiments also demonstrate the robustness of our augmented approach to the quality of the predictions. Our theory crisply quantifies how the theoretical sample complexity is affected by a predictor which satisfies $\|p - \hat{p}\|_{TV} = \alpha$, and correspondingly in practice, we observe that our algorithm is able to take advantage of predictors of varying quality.

Indeed, for the hard instance, we observe that the augmented approach still obtains better error than the standard tester, even if $\|p - \hat{p}\|_{TV}$ is high. This is demonstrated in Figure 5 where we monotonically increase $\beta$ (as described above). The plot shows that while the error also increases monotonically, it is only when $\|p - \hat{p}\|_{TV} \geq .7$ that the quality is comparable to the standard closeness tester.

Indeed, even at high values of $\beta$, $\hat{p}$ is still informative of the heavy-hitter structure of $p$. Therefore, the augmented algorithm can still reliably use $\hat{p}$ to perform a suitable flattening.

The phenomenon repeats itself in for the IP data. In Figure 7(b), we compare the performance of using two different predictors. $\hat{p}_1$ is the same predictor as Figure 7(a) whereas $\hat{p}_2$ is the IP distribution that is in between $p$ and $q$ (it is formed using the chunk of time that is exactly in between the first and the last). We observed that now $\|p - \hat{p}_2\|_{TV} \approx 0.69$ which is much higher than $\|p - \hat{p}_1\|_{TV}$. Nevertheless, it is still the case that many *heavy-hitters* between $p$ and $\hat{p}_2$ are shared. Thus even though the TV distribution between $p$ and the hint $\hat{p}_2$ maybe high, our algorithm can still exploit the shared heavy-hitter structure to obtain improved sample complexity over the un-augmented approach. This demonstrates the practical versatility of our algorithm beyond the precise theoretical bounds.

---
**Algorithm 4** $\ell_2^2$-norm estimation
---
1: **procedure** ESTIMATE-$\ell_2^2$(sample access to $p$, $n$, $\delta$)
2:     **for** $i = 1, \ldots, O(\log(1/\delta))$ **do**
3:         $s \leftarrow \Theta\left(\sqrt{n} \cdot \log(1/\delta)\right)$
4:         $x_1, x_2, \ldots, x_s \leftarrow$ Draw $s$ samples from $p$.
5:         $X \leftarrow \sum_{i=1}^{s} \sum_{j=i+1}^{s} \mathbb{1}_{x_i=x_j}$                $\triangleright$ Number of collisions
6:         $L_i \leftarrow X/\binom{s}{2}$
7:     Output the median of $L_i$'s
---

# F   Background and existing results

## F.1   Estimating the $\ell_2$-norm

In this section, we show that one can estimate the $\ell_2$-norm of a distribution via the number of collisions among $O(\sqrt{n})$ samples. This result was implicitly known from previous work including [52]. Here, we include a formal statement and a proof for the sake of completeness.

**Fact F.1.** *[Adapted from [52]] Suppose we have a parameter $\delta \in (0,1)$ and an arbitrary distribution $p$ over $[n]$ for which we have sample access to. Algorithm 4 receives $n$ and $\delta$ and $s = O\left(\sqrt{n} \cdot \log(1/\delta)\right)$ samples from $p$ and outputs an estimate $L$ of the the $\ell_2^2$-norm of $p$ such that with probability $1 - \delta$, we have:*

$$\frac{\|p\|_2}{2} \leq L \leq \frac{3 \|p\|_2}{2}.$$

*Proof.* Suppose we have $s$ samples drawn from $p$: $x_1, x_2, \ldots, x_s$. Let $X$ denote the number of collisions among these samples. That is, the number of pairs of equal samples $X := \sum_{i<j} \mathbb{1}_{x_i=x_j}$.

In [52], they have shown that $\mathbf{E}[X] = \binom{s}{2} \cdot \|p\|_2^2$ and $\mathbf{Var}[X]$ is at most $2 \left(\mathbf{E}[X]\right)^{3/2}$. Hence, using Chebyshev's inequality, we obtain:

$$\mathbf{Pr}\left[\left|\frac{X}{\binom{s}{2}} - \|p\|_2^2\right| \geq \frac{\|p\|_2^2}{2}\right] \leq \frac{\mathbf{Var}\left[X/\binom{s}{2}\right]}{\|p\|_2^4/4} \leq \frac{8\,\mathbf{E}[X]^{3/2}}{\binom{s}{2}^2 \cdot \|p\|_2^4}$$

$$\leq \frac{8}{\binom{s}{2}^{1/2} \cdot \|p\|_2} \leq \frac{8}{(s/2) \cdot 1/\sqrt{n}} \leq 0.1$$

The second to the last inequality holds since the $\ell_2$-norm of any discrete distribution over $[n]$ is at most $1/\sqrt{n}$. And, the last inequality holds for $s \geq 160\sqrt{n}$. The above inequality implies that each $L_i$ in the algorithm is within the desired bound with probability at least $0.9$.

To find an accurate estimate with a probability $1 - \delta$, we use standard amplification technique: We repeat this process $O(\log(1/\delta))$ times and take the median of these estimates. Using the Chernoff bound, one can show that median is accurate with probability at least $1 - \delta$.

$\square$

## F.2   Standard closeness tester

In this section, we present the result of [30] for robust $\ell_2$-distance estimation between two unknown distributions $p$ and $q$. Their results implies an $\ell_1$-tester for closeness testing of $p$ and $q$ as it has been used in [42]. The important feature of this tester is its adaptive sample complexity to $\max\left(\|p\|_2^2, \|q\|_2^2\right)$ which combined with the flattening technique would achieve the optimal sample complexities for several distribution testing problems. Later [9] have shown that one can modify the this closeness tester such that the sample complexity only deepens on $\min\left(\|p\|_2^2, \|q\|_2^2\right)$.

**Fact F.2** (Adapted from [30, 42, 9]). *Suppose we have two unknown distributions $p$ and $q$ over $[n]$. Let $b \geq \min\left(\|p\|_2^2, \|q\|_2^2\right)$. Algorithm 5 can distinguish whether $p = q$ or $\epsilon$-far from each other with probability $1 - \delta$ using $O(n\sqrt{b}/\epsilon^2)$ samples.*

**Algorithm 5** $\ell_2^2$-norm estimation

---

1: **procedure** STANDARD CLOSENESS TESTER(sample access to $p$, $n$, $\delta$)
2:     $L_p \leftarrow$ ESTIMATE-$\ell_2^2(p,\ n,\ \delta/3)$
3:     $L_q \leftarrow$ ESTIMATE-$\ell_2^2(q,\ n,\ \delta/3)$
4:     **if** $L_p/L_q \notin [1/3, 3]$ **then**
5:         **return** reject
6:     $t \leftarrow O(\log(1/\delta))$
7:     $m \leftarrow O\left(\frac{n \cdot \sqrt{b}}{\epsilon^2}\right)$
8:     accept-count $\leftarrow 0$
9:     **for** $i = 1, \ldots, t$ **do**
10:         Draw $m$ samples from $p$ and $q$.
11:         Let $X_i$ and $Y_i$ denote the number of occurrences of element $i$ in the samples drawn from distributions $p$ and $q$ respectively.
12:         $Z \leftarrow \dfrac{\sqrt{\sum_{i=1}^{n}(X_i - Y_i)^2} - X_i - Y_i}{m}$
13:         **if** $Z \leq \frac{\epsilon}{2\sqrt{n}}$ **then**
14:             accept-count $\leftarrow$ accept-count $+ 1$
15:     **if** accept-count $\geq t/2$ **then**
16:         **return** accept
17:     **return** reject

---

### F.3   Useful facts

**Coupling:**   Suppose we have two probability distributions $p$ and $q$ over a finite domain $\Omega$. A coupling between $p$ and $q$ is a joint distribution over $\Omega^2$ where the two marginals of $\mathcal{C}$ are $p$ and $q$. Below is the fundamental coupling lemma:

**Fact F.3.** *(Coupling Lemma) For every two probability distributions over a finite domain $\Omega$, we have:*

- *For any coupling $\mathcal{C}$ of $p$ and $q$, we have:*

$$\mathbf{Pr}_{(X,Y)\sim\mathcal{C}}[X \neq Y] \geq \|p - q\|_{TV}.$$

- *There exists a maximal coupling $\mathcal{C}^*$ (also known as optimal coupling) for which the above inequality holds:*

$$\mathbf{Pr}_{(X,Y)\sim\mathcal{C}^*}[X \neq Y] = \|p - q\|_{TV}.$$

## G   Augmented flattening for guarantees other than total variation distance

Below, we provide a simple lemma that indicates how one can use $\hat{p}$ to flatten a distribution based on various guarantees of the prediction. Suppose our *target $\ell_2^2$-norm* which we desire is $v$. In the following lemma, we propose a flattening $F$ to decrease the $\ell_2^2$ to $v$ up to some multiplicative factor that depends on the quality of the estimates.

**Lemma G.1.** *Suppose we have an unknown distribution $p = (p_1, p_2, \ldots, p_n)$ over $[n]$. Assume for every $i \in [n]$ we are give an estimate $\hat{p}_i$. Then for every $v \leq 1$, there exists a flattening such that it increases the domain size by $1/v$ and the $\ell_2^2$-norm of the $p^{(F)}$ is bounded by:*

$$\left\|p^{(F)}\right\|_2^2 \leq v \cdot \left(\sum_{i=1}^{n} \frac{p_i^2}{\hat{p}_i}\right).$$

*Proof.* For every $i \in [n]$, we set $m_i$ as follows:

$$m_i := \left\lfloor \frac{\hat{p}_i}{v} \right\rfloor + 1.$$

It is not hard to see:

$$\left\|p^{(F)}\right\|_2^2 = \sum_{i=1}^n \sum_{j=1}^{m_i} \left(\frac{p_i}{m_i}\right)^2 = \sum_{i=1}^n \frac{p_i^2}{m_i} \leq \sum_{i=1}^n \frac{p_i^2}{\hat{p}_i/v} \leq v \cdot \left(\sum_{i=1}^n \frac{p_i^2}{\hat{p}_i}\right)$$

In addition, the new domain size is bounded as follows:

$$\sum_{i=1}^n m_i \leq n + \frac{1}{v} \cdot \sum_{i=1}^n \hat{p}_i$$

Using $\sum_{i=1}^n \hat{p}_i = 1$, we conclude the statement of the lemma. $\qquad\square$

In the following, we discuss a few cases for which we have some guarantee on the accuracy of the estimates and how it affects the $\ell_2$-norm reduction. Note that our desired case here is to not blow up the domain size by more than a constant factor and bring down the $\ell_2$-norm to $O(1/\sqrt{n})$.

1. **Exact values:** Assume $\hat{p}_i = p_i$. By setting $v = 1/n$, the $\ell_2$-norm reduces to $\Theta(1/\sqrt{n})$, and the new domain size is $\Theta(n)$.

2. **Multiplicative upper bounds:** Suppose for every $i$, we are given an estimate $\hat{p}_i$ such that $\hat{p}_i \geq \alpha p_i$ for a parameter $\alpha \geq 1$. Then , we have $\sum_{i=1}^n p_i^2/\hat{p}_i$ is at most $1/\alpha$. Also, let $\beta$ denote $\sum_{i=1}^n \hat{p}_i$. Thus, the sample complexity of the tester in [30], is:

$$\text{\# samples} = O\left(\frac{\text{new domain size} \cdot \min\left(\|p^{(F)}\|_2, \|q^{(F)}\|_2\right)}{\epsilon^2}\right)$$

$$= O\left(\frac{(n + \beta/v) \cdot \sqrt{v/\alpha}}{\epsilon^2}\right) = O\left(\frac{n\sqrt{v/\alpha} + \beta/\sqrt{\alpha v}}{\epsilon^2}\right)$$

$$= O\left(\frac{\sqrt{n\,\beta/\alpha}}{\epsilon^2}\right)$$

Note that the last inequality is achieved by minimizing over $v$ and setting $v = \beta/n$. Moreover, this bound gives us the optimal sample complexity for constant $\alpha$ and $\beta$.

