# OpenReview forum: "Optimal Algorithms for Augmented Testing of Discrete Distributions"
_NeurIPS.cc/2024/Conference — NeurIPS 2024 poster_

### Official Review · Reviewer_qguF · 2024-07-09

**Soundness:** 3
**Presentation:** 2
**Contribution:** 3
**Rating:** 6
**Confidence:** 2

**Summary:**

This paper consider the problem of hypothesis testing for discrete distributions, including identity testing, closeness testing and uniformity testing. The authors investigated these testing problems under the setting where a predicted data distribution is available. Utilizing this additional information, the authors propose "augmented tester", which can reduce the sample complexity when the given predicted distribution is accurate (when it is inaccurate, the proposed algorithm is still robust). Lower bounds are also provided for justifying the optimality of the proposed algorithm.

**Strengths:**

The idea of augmenting hypothesis testing using additional information is interesting. The proposed algorithm can reduce the sample complexity when the given prediction is accurate enough, which is satisfactory. Lower bounds are also provided, making this paper comprehensive.

**Weaknesses:**

A possible weakness is that, the structure of this paper is a little bit weird. You first give your theorems in the introduction when your proposed algorithm is not mentioned yet. Then you provide the proofs in section 2, and state the upper bound again in section 3. The algorithm has not been mentioned until the last.

**Questions:**

I am curious about the intuition behind your algorithm, i.e., how to utilize the additional information.  It will be good if the authors could provide more explanation of the proposed algorithm.

**Limitations:**

no limitations are stated.

---

> ### Author Rebuttal · Authors · 2024-08-07
>
> **presentation**
>
> Thank you for your feedback.  The algorithm is given on page 8 of the main body. We give technical overview in Section 2 before the algorithm to give a high level picture about how our upper bounds and lower bounds (in the appendix) fit together to give a complete characterization of the augmented distribution testing problems we study.
>
>
> **Intuition behind our algorithm**
>
> For the closeness testing problem, we use the hint distribution to $\hat{p}$ to reduce the $\ell_2$ norm of $p$, the unknown distribution. Intuitively, we can do this by looking at the ‘heavy hitters’ of $\hat{p}$: the domain elements that $\hat{p}$ assigns a high probability mass to. Then we split the mass of all heavy elements equally among many synthetically created domain elements. If we see such an element $i$ from our sample from $p$, we perform a thought experiment where we instead sample uniformly from the synthetically created domain elements corresponding to $i$. This does not change any of the TV distances, but significantly reduces the $\ell_2$ norm of $p$ (formalized in Lemma 3.1). A smaller $\ell_2$ norm makes the testing problem significantly easier, as done in prior works. Please see Section 2 for intuition about our other results. For further details, please see Section 2 and Section 3 in the main body.

---

> > ### Comment · Reviewer_qguF · 2024-08-11
> >
> > Thanks for your explanation. I will raise my rating to 6.

---

### Official Review · Reviewer_bNNx · 2024-07-12

**Soundness:** 3
**Presentation:** 3
**Contribution:** 3
**Rating:** 7
**Confidence:** 3

**Summary:**

The paper studies the problem of property testing (specifically uniformity, identity, and closeness testing) in the context of the learning-augmented algorithms framework. They show how a prediction about the underlying distribution could be harnessed to provably reduce the number of samples required when the prediction is of high quality. They give efficient algorithms and provide experimental evaluation with code.

**Strengths:**

The problem is well-motivated and the paper is well written.

**Weaknesses:**

I did not check the proofs in the appendix in detail but the intuition given for the approaches make sense.

**Questions:**

- In the paragraph starting on Line 177: given "perfect predictors", why does one even need additional samples from the distribution? From my limited understanding, that paper talks about having additional access to other kinds of queries on the underlying distribution (apart from just drawing IID samples) and "samples" may refer to those other kinds of queries (for instance, one can query the probability mass of an element and get its true mass in $p$). I'm not sure how that directly compares with your setting where "samples" mean IID samples only.
- How do you obtain an **upper bound** expression for the final big-O containment for the equation block between Lines 378-379? To be precise, we have $s_f \leq = \min\{\frac{n^{2/3} \alpha^{1/3}}{\varepsilon^{4/3}}, n\} \leq \frac{n^{2/3} \alpha^{1/3}}{\varepsilon^{4/3}}$ so $\frac{n}{\varepsilon^2} \cdot \sqrt{\frac{\alpha}{s_f}} \geq \frac{n^{2/3} \alpha^{1/3}}{\varepsilon^{4/3}}$, which is a **lower bound** right? Where did I mess up?

Possible typos:
- Line 266: "...strategy is put the..." should be "...strategy is **to** put the..."?
- First line of the equation between Lines 328-329: Should the last equality be $\leq$?
- Line 375: Should it be $\leq 100 s_f$ instead?
- Line 376: Should it be $\leq 102 n$ instead?

**Limitations:**

Nil

---

> ### Author Rebuttal · Authors · 2024-08-07
>
> **Comparisons to the previous work**
>
> [29] is an example of studying distribution testing when both samples and a prediction distribution $\hat{p} = p$ are available. However, their algorithm does not receive $p$ directly and can only query $p$. Since they do not see the whole $p$, they would require samples from it as well. They consider two types of queries: querying the probability of a certain domain element $p(i)$, or querying the probability of the first $i$ elements for a given $i$. In comparison to our model, their assumption on the accuracy of the prediction is very strong as they assume $p = \hat{p}$. They also consider an alternative model with multiplicative noise, which is still a stronger assumption compared to ours. However, their model is weaker in another sense, as they do not have access to $\hat{p}$ for free. The total cost of their algorithm involves the number of queries they make to $\hat{p}$ and the number of samples (which they describe it as queries to the sampling oracle). For testing uniformity and identity, they have shown that $\Theta(1/\epsilon)$ queries are necessary and sufficient. Given their upper bounds and our lower bounds, it can be concluded that solving the problem in our model is more difficult, meaning it would require many more samples.  Please see Section 1.3 for more detailed comparison to prior related works.
>
>
> **Sample complexity**
>
> For $s_f$, we have two possibilities, either it is equal to $n^{2/3} \alpha^{1/3} / \epsilon^{4/3}$ or it is equal to $n$. For the first case, it is not too hard to see that the equation will be $O(s_f) = O(n^{2/3} \alpha^{1/3} / \epsilon^{4/3})$. Now, assume $s_f$ is $n$. This case, implies that:
> $$n \leq n^{2/3} \alpha^{1/3} / \epsilon^{4/3} \leq n^{2/3}/\epsilon^{4/3}.$$
> Hence, there is an interdependence between the parameters: $n \leq 1/\epsilon^4$. Therefore, one can conclude $n \leq \sqrt{n}/\epsilon^2$. Therefore, both expressions in the sample complexity equate to $O(\sqrt{n}/\epsilon^2)$.
> Maybe it is helpful to note that in this case: $\sqrt{n}/\epsilon^2 \geq n^{2/3}/\epsilon^{4/3} \geq n^{2/3} \alpha^{1/3}/\epsilon^{4/3}$. Hence, our lower bound is not violated.
> [In case we did not answer this question adequately, please leave us a comment, and we will try to clarify further.]

---

> > ### Comment · Reviewer_bNNx · 2024-08-09
> >
> > Thank you very much for clearing up my confusion about the sample complexity!
> >
> > I did read Section 1.3 and I understand that [29] considers a different query model where one can perform stronger queries such as asking for element probabilities $p(i)$. However, from what I understand and what you mentioned, [29] does not get any additional predictor as part of the input. As such, I think it is highly confusing to say that [29] uses "perfect predictors" and I do not think they are directly comparable... In the learning-augmented setup, one can always make a decision by just looking the given predictor without doing any additional work: if the predictor is arbitrarily bad, the answer will just be wrong. Thus, I find it strange to say that an algorithm given perfect predictor even needs any to take any sample.
> >
> > This is still a good piece of work overall and I stick to my positive review. However, as per my response above, I would recommend the authors to reconsider the rephrasing the comparison of [29] in their revision.

---

> > > ### Author Response · Authors · 2024-08-10
> > >
> > > Thank you for your comment. In our future version, we will add the necessary clarification to provide a more comprehensive comparison with [29].

---

### Official Review · Reviewer_Zhut · 2024-07-12

**Soundness:** 3
**Presentation:** 4
**Contribution:** 4
**Rating:** 7
**Confidence:** 1

**Summary:**

The paper looks at the problem of hypothesis testing for discrete distributions where a predicted data distribution is available. The paper gives algorithms (Algorithm 1; closeness testing, Algorithm 3; identity and uniform testing) which either reduces the number of samples required for testing, or do no worse than standard approaches. Lower bounds on samples are given, and experiments are done to validate the performance on real data.

**Strengths:**

The paper is extremely thorough and detailed, giving an overview of the important results (including experiments), as well as a roadmap of the intuition required behind the proofs for the upper bound and lower bounds.

The appendix also includes motivation and explanation for the numerous proofs involved.

The main contributions of needing significantly fewer sample sizes for hypothesis testing compared to CRS'15 or [41] is also significant.

**Weaknesses:**

I am not sure whether this is an appropriate venue mostly due to the heavy technical details in the paper, and with a lot of key details in the appendix ; but to be fair, the conference format makes it difficult to present detailed work like this.

Unfortunately, this is not mostly in my area of speciality, so I am unable to give meaningful comments about potential weaknesses (or strengths).

**Questions:**

These questions may be straightforward to those familiar with the relevant literature, but these are the questions I have:
 - Are there certain types of data where making weak assumptions about the predictor (compared to related works) might give bad results (or similar results to existing methods)?
 - Is there a reason why experiments for identity testing and uniformity testing are not given?

---

> ### Author Rebuttal · Authors · 2024-08-07
>
> **Suitability for the conference**
>
> We would like to point out that many related works on distribution testing have appeared in recent ML conferences such as NeruIPS/ICML/ICLR. These works are relevant to the learning-theory, privacy, algorithmic statistics, and sublinear algorithms community within ML. Please see below for a small sample of such related works that appeared in recent NeurIPS conferences that we cite in our work. Our references in our submission list additional related works in other top ML venues.
> [4] J. Acharya, Z. Sun, and H. Zhang. Differentially private testing of identity and closeness of
> discrete distributions. NeurIPS 2018
>
> [5] Jayadev Acharya, Constantinos Daskalakis, and Gautam Kamath. Optimal testing for properties of distributions. NeurIPS 2015
>
>  [8] Maryam Aliakbarpour, Mark Bun, and Adam Smith. Hypothesis selection with memory
> constraints. NeurIPS 2023
> [9] Maryam Aliakbarpour, Ilias Diakonikolas, Daniel Kane, and Ronitt Rubinfeld. Private testing of distributions via sample permutations.  NeurIPS 2019
>
> [24] Clement L. Canonne, Ilias Diakonikolas, Daniel Kane, and Sihan Liu. Nearly-tight bounds for testing histogram distributions. NeurIPS 2022
>
> [27] Clement L Canonne, Gautam Kamath, Audra McMillan, Jonathan Ullman, and Lydia Zakynthinou. Private identity testing for high-dimensional distributions. NeurIPS 2020
>
> [45] Ilias Diakonikolas, Daniel M. Kane, and Alistair Stewart. Sharp bounds for generalized uniformity testing. NeurIPS 2018
>
>
> **Weaker assumptions about prediction**
>
> Thank you for suggesting an interesting open direction! It is not clear to us if our predictor assumption can be relaxed while retaining similar guarantees. We believe our model is quite natural, but there could certainly be other weaker assumptions which are powerful enough to give improved sample complexity. However, we note that our algorithms are robust, so even if the predictor assumptions do not hold, we never lose asymptotically on the sample complexity compared to classical algorithms without predictions.
>
>
> **Experiments on identity testing and uniformity testing**
>
> We focused on closeness testing since it is the hardest statistical task out of the three distribution testing problems we studied: it generalizes identity and uniformity testing, and a larger sample complexity is required, both in the classical and augmented settings. We suspect that one can see similar qualitative gains from using our augmented algorithms, provided that appropriate predictions are available.

---

> > ### Comment · Reviewer_Zhut · 2024-08-07
> >
> > Thank you for the detailed rebuttal. My score remains unchanged.

---

### Official Review · Reviewer_BZRn · 2024-07-12

**Soundness:** 3
**Presentation:** 3
**Contribution:** 3
**Rating:** 7
**Confidence:** 4

**Summary:**

This paper studies the task of identity/closeness testing when the tester is augmented with predictions of the unknown distribution involved apriori. In particular, in identity testing, in addition to sample access to the unknown distribution $p$, the algorithm is also given some $\hat p$ that may or may not satisfy the guarantee $ \text{TV}(p, \hat p) \leq \alpha$. If the guarantee does not hold, the algorithm is allowed to answer **inaccurate**. Otherwise, the algorithm is asked to perform the standard testing task: return YES if $p$ equals to some known distribution $q$, and NO if $p$ is at least $\epsilon$-far from $q$ in total variation distance. For closeness testing, though both $p$ and $q$ are unknown, the authors assume that such prior prediction $\hat p$ is only available for one of the distributions.

For identity testing, the sample complexity goes through a phase transition depending on the relative size of the prediction error tolerance parameter $\alpha$ and $d = \text{TV}(q, \hat p )$, the TV distance between the prediction distribution and the known distribution. When $d<\alpha$, the sample complexity is the same as the usual identity testing. When $d > \alpha$, the sample complexity is $\min\left( \frac{1}{ (d - \alpha)^2 }, \frac{\sqrt{n}}{\epsilon^2} \right)$. Intuitively, this is saying that the prediction could potentially help a lot if it predicts that the test is in the soundness case (as $\hat p$ is indeed far from $q$) but not so much if it predicts the test is in the completeness case. This intuition is clear from their upper bound approach. They leverages the Scheffé set $S$ (the set that realizes the maximum discrepancy between $\hat p$ and $q$), and test for discrepancy between $p(S)$, $q(S)$ and $\hat p(S)$. This will either invalidate the prediction or lead to rejection. Furthermore, since this is simply a 1-dimensional bias estimation problem, they could avoid any dependency on the doamin size (when $d$ is sufficiently separated from $\alpha$). The above strategy is not so helpful when there is no significant discrepancy between $p(S)$, $q(S)$ and $\hat p(S)$, and therefore their algorithm fall back to the standard identity tester when the prediction says that the test is in the completeness case.

For closeness testing, they show that the sample complexity is given by $\sqrt{n} \alpha^{1/3} / \epsilon^{4/3} + \sqrt{n} / \epsilon^2$. Here the Scheffé set strategy no longer applies as $q$ is also unknown to us. However, the authors show that the prior $\hat p$ is still very useful in an important closeness testing sub-routine (commonly referred as **flattening**). At a high level, when the unknown distributions have large $\ell_2$ norm, standard collision-based test statistics may have large variance. Flattening techniques could then be used to transform the distributions to reduce their $\ell_2$ norms, and hence also the variance of the test statistics. The authors show that such a routine can be implemented in a more sample-efficient way if the algorithm is equipped with a prediction $\hat p$.

**Strengths:**

Identity/closeness testings are fundamental problems in many different areas. In the standard setting, the sample complexity scales polynomially with respect to the domain size, making the tests prohibitively expensive for distributions with large supports. The authors demonstrated that the sample complexity may be significantly improved if the tester is augmented with a prediction of the unknown distribution. The surprising part is that the prediction need not to be an accurate one. While the tester draws significantly fewer samples if the prediction is accurate, even if the prediction is not, the tester is guaranteed not to be misguided. In particular, it can identify inaccuracies in the prediction, and simply fall back to the standard testing approach in that case. Lastly, the bounds are optimal up to constant factors and technically solid.

**Weaknesses:**

The authors mention that the potential application is when the unknown distribution evolves over time. In that case, the ``unknown'' distribution is not completely unknown to us as we may extract information from past data. It will be more compelling if the authors could provide more detailed mathematical setup of this. For example, it will be interesting to see analysis of the behavior of augmented testers when we face a sequence of tests where the unknown distribution may go through random distribution shifts from one task to another.

**Questions:**

Have the authors considered situation when we have prior knowledge about both of the unknown distributions in closeness testing? Then it seems like the Scheffé set may again become useful, and could lead to huge sample complexity improvements.

Comment: The setup bears some similarities to the notion of testable learning, where the algorithm is given some prior distribution assumption that may or may not hold. In particular, the algorithm is also given the ability to output ``reject'' when this prior knowledge is false.

**Limitations:**

Yes, they have.

---

> ### Author Rebuttal · Authors · 2024-08-07
>
> **Clarification on evolving distributions**
>
> For our results, the algorithm requires access to a prediction distribution $\hat{p}$, which is intended to be close to the unknown distribution $p$ (in addition to samples from $p$). Our algorithms are applicable in settings where the extra information available about $p$ can be translated into a distribution $\hat{p}$. In our introduction, we strive to depict various scenarios where such predictions can be available, including slowly evolving distributions such as network traffic data or search engine queries. Mathematically, we consider a series of slowly changing unknown distributions: $p_1, p_2, \ldots, p_t$. In this setting, the empirical distribution of the samples from $p_1, p_2, \ldots, p_{t-1}$ can serve as a prediction for $p_t$. For example, the distributions of search queries change slightly every hour or so. However, the overall empirical distribution of search queries from the past year could be a good prediction for the distribution at a particular hour. We will clarify this in our introduction. Continual testing while the distributions undergo small perturbations at every step is a very interesting open question. Although our results do not directly address this problem, we believe that some techniques from augmented flattening could potentially be helpful in solving this problem as well.
>
>
> **Studying two predictions**
>
> Thank you for suggesting an interesting open direction. We did not consider this case in our paper. First, we would like to mention that our upper bound still holds when two predictions are provided. It is not too difficult to adapt our augmented flattening to incorporate both $\hat{p}$ and $\hat{q}$. The sample complexity in this setting is proportional to the minimum of $\alpha_p^{1/3}$ and $\alpha_q^{1/3}$. However, the lower bounds do not hold. As you pointed out, one can hope to achieve better sample complexity if the distance between the two predictions is more than $\alpha_p + \alpha_q + \epsilon$. In particular, estimating the probability of the Scheffé set gives us either a 'reject' or 'inaccurate information' result for one of the distributions. On the other hand, when $\hat{p}$ and $\hat{q}$ are identical, our lower bound indicates that one cannot hope for better sample complexity. It remains an open question to study the case where the distance between the two predictions is smaller than $\alpha_p + \alpha_q + \epsilon$, but larger than zero.
>
>
> **Connection to testable learning**
>
> Yes, in both cases, the algorithm is given an option of forgoing to solve the problem when the underlying assumption does not hold. Although there is no notion of ``augmentation’’ in that line of work.

---

### Author Rebuttal · Authors · 2024-08-07

We thank all the reviewers for their feedback. We will integrate all your editorial comments regarding the presentation of our paper in the future version.

---

### Decision · Program_Chairs · 2024-09-25

**Decision:**

Accept (poster)

**Comment:**

This paper studies the problem of identity/closeness testing when the tester is augmented with predictions of the unknown distribution involved apriori. Under this setting, the paper presents sample complexity bounds for uniformity, identity, and closeness testing problems and shows them to be smaller than that of their classical counterparts. Moreover, lower bounds are presented to establish the optimality. All the reviewers are positive about the paper's novelty and contributions, which I agree with. Based on the discussion with the reviewers, I think the paper is a nice contribution to the property testing literature and therefore I am recommending acceptance. I strongly urge the authors to incorporate all the reviewer comments along with any clarifications they provided in the rebuttal into the camera-ready version.